

# Free–Surface Flow as a Variational Inequality (*evolve_glacier v1.1*): Numerical Aspects of a Glaciological Application

Anna Wirbel[1] and Alexander Helmut Jarosch[2]

[1]Department of Atmospheric and Cryospheric Sciences, Universität Innsbruck, Innsbruck, Austria
[2]Independent Researcher, Hörfarterstrasse 14, Kufstein, Austria.

**Correspondence:** Anna Wirbel (Anna.Wirbel@uibk.ac.at)

**Abstract.** Like any gravitationally driven flow that is not constrained at the upper surface, glaciers and ice sheets feature a free–surface, which becomes a free boundary problem within simulations. A kinematic boundary condition is often used to describe the evolution of this free–surface. However, in the case of glaciers and ice sheets, the naturally occurring constraint that the ice surface elevation (S) can not fall below the bed topography (B), $(S - B \geq 0)$ in combination with a non–zero mass balance rate

complicates the matter substantially. We present an open–source numerical simulation framework to simulate the free–surface evolution of glaciers that directly incorporates this natural constraint. It is based on the finite element software package FEniCS solving the Stokes equations for ice flow and a suitable transport equation, i.e. 'kinematic boundary condition', for the free–surface evolution. The evolution of the free–surface is treated as a variational inequality, constrained by the bedrock underlying the glacier or the topography of the surrounding ground. To solve this problem, the 'constrained' non–linear problem solving

capabilities of PETSc's SNES interface are used. As the constraint is considered in the solving process, this approach does not require any *ad–hoc* post–processing steps to enforce no–negativity of ice thickness as well as mass conservation. The simulation framework provides the possibility to partition the computational domain so that individual forms of the relevant equations can be solved for different subdomains all at once. In the presented setup, this is used to distinguish between glacierized and ice–free regions. The option to chose different time discretizations, spatial stabilisation schemes and adaptive mesh refinement

make it a versatile tool for glaciological applications.

We present a set of benchmark tests that highlight the simulation framework is able to reproduce the free–surface evolution of complex geometries under different conditions for which it is mass conserving and numerically stable. Real–world glacier examples demonstrate high resolution change in glacier geometry due to fully–resolved 3D velocities and spatially variable mass balance rate, whereby realistic glacier recession and advance states can be simulated. Additionally, we provide a thorough

analysis of different spatial stabilisation techniques as well as time discretization methods. We discuss their applicability and suitability for different glaciological applications.

## 1 Introduction

Free boundary problems arise naturally in geophysics. For these kind of problems, in addition to the solution function, parts of the domain itself, specifically the free boundary, are also unknown. Gravitationally driven fluid flows common in geophysics





(e.g. water, ice, lava) that are not constrained from the above (e.g. White, 2010) are examples of such free boundary problems. In addition, melting–freezing processes such as the two–phase Stefan problem (e.g Alexiades, 1992), or marine ice–sheet grounding line dynamics (e.g. Schoof, 2011; Goldberg et al., 2018) contain free boundaries as well. Many free boundary problems can be seen and analyzed as variational inequalities (e.g. Kinderlehrer and Stampacchia, 1980), where modern numerical

methods and software tools facilitate their solution.

In this paper, we focus on the dynamics of ice, modelled as a non–linear viscous gravity–driven flow, which, due to its free–surface nature, forms a free boundary problem. Ice dynamics as free boundary problems have been studied before, although often with a reduced stress balance known as the shallow ice approximation (SIA, Mahaffy, 1976; Hutter, 1983). A mathematical analysis of SIA flow, formulated as an obstacle problem (a classical example for free boundary problems) was carried

out by Jouvet and Bueler (2012). The use of finite–difference methods in combination with suitable flux–limiting schemes proved to be successful in solving the margin and free–surface projection step within SIA ice flow (Jarosch et al., 2013). In the vertically integrated ice flow model Úa (Gudmundsson, 2019), which utilizes finite elements to solve for SIA and shallow shelf approximation (SSA) ice flow (e.g. Gudmundsson et al., 2012), Lagrange multipliers (e.g Ito and Kunisch, 2008) operating on the momentum equation are used to implement a constrained free–surface. More recently, a different numerical analysis

proposed a mixed finite–volume–element approach for solving SIA ice flow as a variational inequality (Bueler, 2016).

Few existing numerical models have documented variational inequality capabilities to solve free boundary problems that consider the full stress balance (i.e. Stokes flow) in ice dynamics. To our knowledge, only Elmer/Ice recognizes the variational inequality nature of the free–surface in such ice flows (e.g. Zwinger and Moore, 2009). Elmer/Ice imposes Dirichlet conditions on the constrained boundaries that are iteratively released by a criterion based on residuals (Gagliardini et al., 2013), whereas

instead we here utilize reduced–space methods (Benson and Munson, 2006). Our novel approach combines an existing finite–element Stokes flow model (Jarosch, 2008; Wirbel et al., 2018) with highly efficient, existing variational inequality solving numerical libraries (PETSc, Balay et al., 2018a, b, 1997) that have been successfully applied to SIA ice flow (Bueler, 2016). Thus we create an efficient, flexible and ready–to–use simulation framework for the evolution of land–terminating ice bodies. In this paper, we refer to glaciers for simplicity, but this approach and results are equally applicable to other land–terminating

ice bodies such as ice sheets. For evaluating our scheme, we propose a new set of free–surface evolution benchmarks that will be useful tests for other existing or future implementations. In addition, we review the following stabilization schemes: (1) Streamline Upwind Petrov–Galerkin (SUPG), (2) Continuous Interior Penalty (CIP, Burman and Hansbo (2004)), (3) Spurious Oscillations at Layer Diminishing (SOLD, John and Knobloch (2008)), (4) 4th–order bubble–enriched functions (BR, e.g. Brezzi et al. (1992)) and (5) discontinuous Galerkin with upwinding (DG). We also investigate the effect of applying adaptive

mesh refinement on numerical stability. In combination with the following time discretization schemes: (1) Crank-Nicholson, (2) Backward-Euler and (3) second order Runge Kutta (Gottlieb and Shu, 1998), we discuss their suitability for glaciological applications.

The setup presented here has been developed to be fully compatible with an existing glacial debris transport model *debadvect* (Wirbel, 2018), with the wider aim of developing a full debris–covered glacier system model (Wirbel et al., 2018). Nevertheless

our treatment of the free–surface evolution is widely applicable to geophysical flows.





This paper is organized as follows. First we review the mathematics of free–surface ice flows and how they form free boundary problems. Thereafter we provide details of the numerical methods and the model chain of our approach, describe the results of standardized benchmark tests of our free–surface advection scheme and present three different applications for real–world glacier geometries. Following that, we discuss general stability issues of finite element methods for the simulation

of the free–surface evolution of glaciers, and analyze the results of using the different stabilization approaches in combination with the different time discretization schemes. The manuscript closes with a discussion of results and a conclusion.

## 2 Mathematical Formulation

We study the evolution of glaciers within a spatial domain $\Omega \, \epsilon \, \mathbb{R}^3$, where the ice–air boundary ($\partial\Omega_{\mathrm{top}}$) is of special interest. Within $\Omega$ we track two elevation fields, the ice surface elevation ($S$) and the bedrock elevation ($B$). Glaciers exist wherever

$S > B$ within $\Omega$, and elsewhere the landscape is considered to be free of ice ($S = B$). The natural constraint that the ice surface elevation can not fall below the bedrock elevation:

$$S \geq B, \tag{1}$$

must be fulfilled at every point in time and space within $\Omega$.

Velocities for the slow, gravity–driven flow of ice are computed with the stationary incompressible Stokes equations (see

Wirbel et al. (2018) for details). We treat the surface of a glacier as stress–free for which:

$$2\eta\boldsymbol{\varepsilon} \cdot \mathbf{n} - p\mathbf{n} = 0 \quad \text{on } \partial\Omega_{\mathrm{top}}. \tag{2}$$

Here $\mathbf{n}$ is the outward pointing surface normal, $\boldsymbol{\varepsilon} = 1/2(\nabla\mathbf{u} + (\nabla\mathbf{u})^T)$ is the strain rate tensor and $p$ is the pressure. In order to describe the evolution of this "free" surface as a consequence of ice motion and specific mass balance rate, we employ the following advection equation:

$$\frac{\partial S}{\partial t} = -u_h \cdot \nabla_h S + u_z + \dot{a}, \tag{3}$$

where $u_h = (u_x, u_y)$ are the horizontal ice surface velocity components, $u_z$ is the vertical ice surface velocity component and $\dot{a}$ the specific surface mass balance rate in $\mathrm{ms}^{-1}$. Eq. 3, known as a "kinematic boundary condition" in fluid dynamics (e.g. White, 2010), is here shaped into a glaciological context by including the specific mass balance rate (e.g. Hutter, 1983).

Due to the constraint Eq. 1 puts on Eq. 3, the free–surface evolution of glaciers becomes a variational inequality (Kinderlehrer

and Stampacchia, 1980), previously realized by e.g. Zwinger and Moore (2009); Jouvet and Bueler (2012).

## 3 Numerical treatment

We apply a Finite Element Method (FEM) approach, implemented in Python and using the software packages FEniCS (https://fenicsproject.org, Alnæs et al., 2015; Logg et al., 2012) and PETSc (Balay et al., 2018a, b, 1997) for solving Eq. 3.





Ice velocities are computed on a three–dimensional (3D) mesh using *icetools*, an open–source full–Stokes model for ice flow (Jarosch, 2008; Wirbel et al., 2018). As the free–surface evolves on $\partial\Omega_{\text{top}}$, the problem has one dimension less than the ice flow problem itself (defined on $\Omega$). The free–surface evolution is thus solved on a two–dimensional (2D) mesh with the same horizontal extent as the 3D mesh. On this 2D mesh, the surface elevation is defined as a continuous function on piecewise

linear elements. There are three different options for the time discretization that can be chosen to best suit the characteristics of the problem at hand.

Firstly, we provide a semi–implicit Crank–Nicholson scheme (see Appendix B1), where we apply linearization by using $u_h$, $u_z$ and $\dot{a}$ from the previous time step instead of considering their actual non–linear dependence. This results in a weak coupling of free–surface evolution and velocity computations. For this setup, extensive simulation tests show that very conservative time

stepping is preferable. Hence, we derive the time step with the Courant-Friedrich-Lewy (CFL) condition, using the maximum velocity, i.e. the maximum of horizontal or vertical velocity or specific mass balance rate and a Courant number ($c_{max}$) of 0.1. Secondly, a Backward-Euler time discretization (see Appendix B2) can be chosen, where, again due to linearization, the velocity and mass balance rate fields are still related to the resulting surface elevation of the previous time step. In this case, Eq. 3 is only solved once per surface evolution computation. This option is offered to provide a more robust procedure

regarding numerical stability, however this comes at the cost of additional smoothing (for details on stability properties see Sect. 7). Thirdly, we provide a total variation diminishing time discretization, a fully explicit Runge–Kutta (RK2) scheme of second order (Gottlieb and Shu, 1998) (see Appendix C for a detailed description). In this case, the coupling between velocity computations and free-surface evolution strongly increases, which is particularly favourable for glacier simulations in an advancing state (see Sect. 7 for further information), but greatly increases computational costs due to the additional

velocity computations required. This scheme, in combination with shock–capturing artificial viscosity and the SUPG method for spatial stabilisation, has been employed in VarGlas, a higher-order ice sheet model (Brinkerhoff and Johnson, 2013), to ensure stability.

For the Crank–Nicholson- and Backward Euler time discretizations, the coupling, i.e. iterative computations of glacier velocity and the free–surface evolution is determined by the surface evolution time step. Firstly, an update of the computational

mesh is required after a significant change in surface geometry has occurred and ice velocities have to be computed for the new geometry in order to account for its impact on ice flow. This is important to ensure consistency of velocity fields and glacier geometry. The update interval should be chosen to suit the respective study case. For example, glaciers with fast moving regions might require a more frequent mesh update to accurately represent the changes in geometry and interactions with ice flow. Secondly, the surface evolution time step has to be chosen according to the mass balance rate conditions represented,

i.e. dependent upon whether mass is added and removed in a continuous manner over longer periods or in a single event of mass change. For example, to represent constant additions and loss of mass with time, frequent geometry updates and hence, velocity field computations are required. In a glaciological context, it is tempting to chose this update to take place in yearly intervals in coherence with the availability of surface mass balance estimates typically generated for each hydrological year. However, it is crucial to chose a suitable coupling time stepping of velocity field computations and free–surface evolution in





order to correctly represent the mass balance rate aimed for. If a Runge–Kutta time discretization is chosen, stronger coupling is inherent to the method.

The spatial derivatives are discretized using a standard Galerkin FEM and for stabilization we apply the SUPG approach. This is done by adding an additional term to the variational form of Eq. 3, which is based on its residual and a mesh size dependent

stabilization parameter. We define this parameter of $\mathcal{O}(h_K)$ as $\tau = \frac{h_K}{2||\mathbf{u}||}$ (Bochev et al., 2004; John and Novo, 2011), where $h_K$ is a measure of the local cell size and $\mathbf{u}$ is the divergence-free velocity field. This stabilization scheme has been established for advection–diffusion equations in the advection–dominated case (i.e. Peclet numbers greater than 3) and is also suitable for our advection problem. For the Runge–Kutta scheme, the SUPG stabilization is combined with a Spurious Oscillations at Layers Diminishing (SOLD, John and Knobloch (2008)) method. Further information on these spatial stabilization schemes and on

the issue of how to stabilize FEM for advection problems with respect to glaciological applications in general, is provided in Sect. 7.

## 4 Model chain

The free–surface evolution is computed in several steps utilizing different software packages and is now provided as a fully–automated simulation framework.

The specific steps which are performed to simulate the free–surface evolution are: (1) preprocessing and mesh generation, (2) solving free–surface evolution and update STL (stereolithograph) [1] or *msh* file (gmsh's native file format), (3) generate a new mesh and optional mesh modifications.

### 4.1 Preprocessing and mesh generation

The computations are performed on meshes, built from triangles in 2D and tetrahedrons in 3D. These are created with *gmsh*

(Geuzaine and Remacle, 2009), an open–source finite element mesh generator. In case of adaptive mesh refinement we employ gmsh's capabilities to introduce attractor fields to create regions of different mesh size.

In the preprocessing step, a 3D mesh is generated from a digital elevation model (DEM) using gmsh software (Geuzaine and Remacle, 2009). The DEM covers the glacier, or multiple glaciers of interest and their surrounding terrain, including information on the glacier bed geometry. To allow glacier geometry change and localized mesh refinement we introduce this

versatile algorithm.

First, a rectangular domain is defined covering the area of interest so that potential glacier advances as well as processes that require information on the surrounding terrain (e.g. gravitational mass additions from surrounding slopes) can be represented correctly. Its corner coordinates are used to create a STL or msh file for a 3D rectangular box covering the domain of interest. For the following steps, two options are available, either operations are performed on the stl file or the msh file directly. If the

msh file is used, no remeshing is required but the number of vertical layers are fixed and the mesh becomes structured in the

---

[1]STL files describe 3D surfaces with triangles defined by their vertices and facet normals. This offers the possibility to modify the vertex coordinates directly, and thereby change the surface elevation.





vertical. If the stl file is used, additional vertical elements might be introduced by the meshing software, in case of significantly increased ice thickness. The default setting is to use the stl files.

In a next step, the DEM is used to directly set the vertical coordinates of the surface and bed elevation at the vertices of this box surface STL or box msh file. This updated file is then used to create a 3D volume mesh.

A 3D mesh is required for the velocity computations, even where the ice thickness ($H = S - B$) is zero. In those regions a minimal thickness $h_{af}$ is assigned, which we refer to as the "artificial ice layer" for ease of understanding.

### 4.2    Solving Free–Surface Evolution

In this step, the evolution of the free–surface is computed by solving Eq. 3. The solution is then used to modify the vertical coordinates of the box surface STL or box msh file accordingly, in order to produce an updated 3D glacier geometry mesh.

The free–surface evolution itself is computed using the relevant terms and quantities defined on a 2D computational mesh. The following procedure offers a flexible setup that works for any given mesh size in an automatized manner to initially derive all quantities on the 2D mesh. This step requires some mesh modifications as well as the redefinition of vector and scalar functions, which is described in detail in the following paragraph and illustrated in Fig. 1.

First, a boundary mesh of the initial 3D geometry (Fig. 1 (1)) is created and from this, submeshes of the surface and bed are

generated with marked surface and bed boundaries (Fig. 1 (3)). On these submeshes, which are 2D surfaces but still oriented in 3D space, a function for the actual surface elevation, i.e. the vertical coordinate of the submesh, is introduced and the 3D ice velocities (see Fig. 1 (2)) are interpolated onto the submesh of the surface. The z-coordinates of all the submeshes are then set to zero. On these now flat meshes, the elevation of the surface and bed, which was the actual vertical coordinate of the submeshes, are still well defined. This step allows us to evaluate the surface elevations with only the horizontal coordinates as

input.

Equation 3 is then solved on a 2D computational mesh (see Fig. 1 (4)) covering the horizontal extent of the 3D glacier geometry mesh (see Fig. 1 (1)). If gridpoint locations are not kept constant throughout the simulations, topographical features such as e.g. ridges, could become smoothed if the gridpoints of highest or lowest locations are not represented in the modified mesh, even where there is no change due to mass balance rate or ice flow. Hence, if adaptive mesh refinement is used, refinement

locations have to include regions where important topographic features are located also outside of the glacierized domain.

The functions representing the elevation of surface and bed, as well as the 3D velocity field, are evaluated at the vertices of the 2D mesh (see Fig. 1 (4)). In Eq. 3, the horizontal and vertical velocity component are required as separate inputs, hence these individual components are defined as individual vector and scalar functions (see Fig. 1 (4) labelled with xy and z).

Following the problem definition of the free–surface evolution as a variational inequality, the possible solutions of the surface

elevation $S$ belong to a complex set, in other words, the surface elevation $S$ cannot fall below a constraint, which is the glacier bed elevation $B$. As we introduced an artificial ice layer of thickness $h_{af}$ to construct a 3D volume mesh over the entire domain, this constraint is defined as $\tilde{B} = B + h_{af}$. There is no actual upper constraint, so we set it to a value above the highest present surface elevation and possible additions according to the mass balance rate.





In order to eliminate the effect of the artificial ice layer on the free–surface evolution, we introduce subdomains separating ice–covered and ice–free areas of the computational domain. In subdomain 1, where there is actual ice ($S > \tilde{B}$), the full form of Eq. 3 is solved, whereas in subdomain 2, where no ice is present ($S = \tilde{B}$), Eq. 3 reduces to:

$$\frac{\partial S}{\partial t} = \dot{a}. \tag{4}$$

These subdomains are defined according to ice thickness. In order to allow the glacier to advance into previously ice-free areas, subdomain 1 is enlarged by a velocity–dependent buffer zone. Additionally, velocities within the artificial ice layer are set to zero.

    In order to deal with the variational inequality characteristic of Eq. 3, we employ the "constrained" non–linear problem solving capabilities of PETSc's SNES interface via the FEniCS framework. Our standard approach is to use reduced–space

methods as these proved faster convergence in our tests (see Sect. 5), however semi–smooth methods are available as well (Benson and Munson, 2006). To get a decent initial guess for the computations, we first solve the non–linear problem using the Newton method without any constraints. This step is not only relevant to make computations at all possible for complex cases, but also leads to a significant speed up of the solving process.

    As a result of these computations, the updated surface elevation is used to modify the vertical coordinate in the box surface

STL file (or the box msh file) including the glacier surface, bed and the surrounding topography (see Fig. 1 (5)).

### 4.3   Mesh update

In this step, the updated domain STL is used to mesh a volume using the open-source finite element mesh generator gmsh (Geuzaine and Remacle, 2009). If the msh file is updated, thereby a new mesh is generated directly without requiring any remeshing.

## 5   Simulation Framework Tests

Firstly, we provide a set of tests that we refer to as "benchmark tests". These evaluate the simulation framework performance with demanding problems addressing the robustness of our numerical implementation. For these tests, known solutions exist which allow us to perform a quantitative error assessment of the simulation framework. Secondly, we demonstrate the simulation framework capabilities for a glacier case by simulating the free–surface evolution of an alpine valley glacier for different

mass balance conditions, assuming no sliding at the glacier bed. In addition, we provide a third glacier case, where we introduce extremely strong gradients through randomly generated input fields of velocity and mass balance rate. All of the tests in this Section are performed with a Crank–Nicholson time discretization and the computational time step is derived with the CFL condition using respective maximum values of velocity and mass balance rate and a Courant number of 0.1. In the benchmark tests the artificial ice layer thickness is set to 0.5 m and in the glacier tests it is set to 1 m.



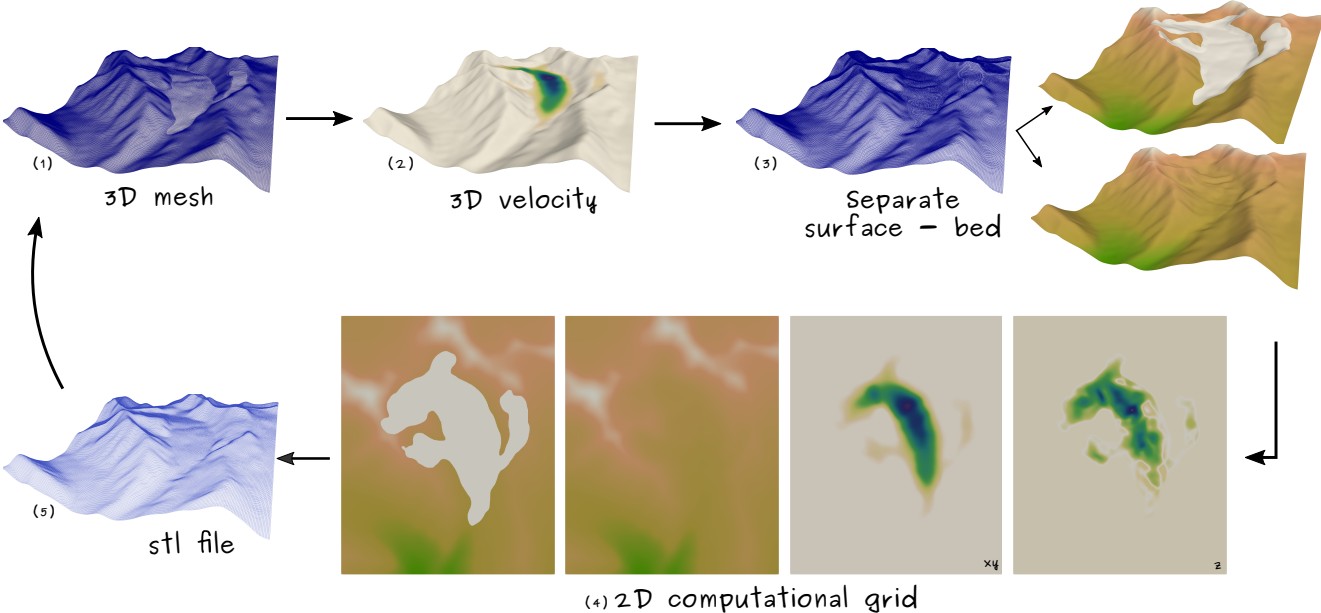

**Figure 1.** Workflow of step 2. (1) 3D mesh of the glacier including its surroundings, (2) 3D velocity field computed with *icetools*, (3) boundary mesh of the 3D geometry and submeshes of surface and bed boundary, (4) surface elevation, bed constraint, horizontal and vertical velocity as functions defined on 2D computational mesh and (5) updated STL file used to create new 3D mesh.

## 5.1 Benchmark tests

The simulation framework performance is evaluated in terms of its capability to reproduce analytical results, conserve mass, be numerically stable and converge as a function of spatial resolution. The tests are not performed for geometries in line with glaciological scales, but are appropriate to test the performance of the simulation framework.

5    Regions of steep gradients are most demanding for the simulation framework to produce accurate and numerically stable solutions. Thus, we choose a pyramid–shaped geometry to test its performance. Our initial 3D geometry of the free–surface problem is formed by a flat plane ($10 \times 10$ m$^2$) and an irregularly shaped pyramid on top of $1$ m height, as shown in Fig. 2.

Two different velocity fields and mass balance rate representations are used to test the simulation framework capabilities for a wide range of settings. In Test A, the prescribed velocity field introduces a translation (indicated by blue arrow in Fig. 2) of

10  the pyramid including a vertical displacement due to the effect of a prescribed vertical velocity and mass balance rate field. In Test B, the pyramid undergoes a forward and reversed swirling flow rotation (indicated by green arrow in Fig. 2) following the test described in LeVeque (1996).

In this manner, we test the simulation framework by introducing 3D displacement including overall negative mass changes over time as well as complex, spatially and temporally variable velocity fields that induce strong distortion of surface geometry.



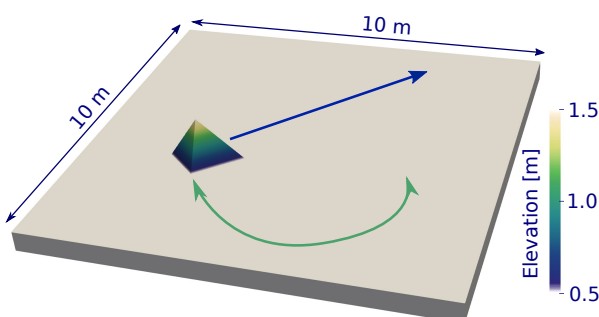

**Figure 2.** Initial 3D geometry for benchmark tests with surface elevation in colour and arrows indicating the direction of movement for Test A in blue and for Test B in green.

### 5.1.1 Test A

A translational movement is prescribed by a velocity field of $u_{\mathrm{x}} = 0.85$ ms$^{-1}$ in x–direction, $u_{\mathrm{y}} = 0.55$ ms$^{-1}$ in y–direction and $u_{\mathrm{z}} = 0.15$ ms$^{-1}$ in z–direction, whereas the mass balance rate is given by $\dot{a} = -0.30$ ms$^{-1}$. This creates a horizontal movement of the test pyramid with an effective vertical downward motion, thus the pyramids volume will decrease over time. The computations are performed on structured meshes for four different mesh resolutions of $N = 125, 250, 500, 1000$, corresponding to $dx = dy = 0.08, 0.04, 0.02, 0.01$ m. For the free–surface computations, a Dirichlet boundary is set so that $S = S_0$ on the entire domain boundary and the computational time step is derived using the CFL condition and a Courant number of $c_{max} = 0.1$. A new mesh geometry is created every $0.5$ s and simulations evolve until $6.5$ s, when the pyramid has almost vanished from the computational domain.

The initial geometry displayed in Fig. 2 has a volume of $V_{\mathrm{init}} = 0.37\dot{6}$ m$^3$. An exact solution of the volume change for the pyramid over time can be derived using the following set of equations. Starting with the initial pyramid base area $A_{\mathrm{init}} = 1.13$ m$^2$, its temporal change for time $t$ is described as

$$A_{\mathrm{base}}(t) = A_{\mathrm{init}} \left( \frac{h + z(t)}{h} \right)^2, \tag{5}$$

with $h = 1$ m the initial pyramid height and $z(t) = (u_z + \dot{a})t$ the effective vertical displacement over time. In our advection case, the horizontal velocities do not have any influence on the pyramid volume. Having derived the changing base area (Eq. 5), we can simply calculate the changing pyramid volume $V$ such that

$$V(t) = \frac{1}{3} A_{\mathrm{base}}(t) \left( h + z(t) \right). \tag{6}$$

This exact solution for Test A will be used in Sect. 6.1.1 to evaluate the simulation framework performance for different mesh sizes and to demonstrate convergence.





### 5.1.2 Test B

For this test, a time dependent velocity field is prescribed. It induces a swirling flow following the velocity field of Example 9.5 in LeVeque (1996). In the original test, three features of different shape are advected. Here we apply an identical velocity field, but using the pyramid geometry shown in Fig. 2. Due to the prescribed surface velocity, the shape of the pyramid is

altered drastically during the test, however the shape will be recovered due to the reverse flow field at the end of the simulation. Comparing initial and final geometry forms a measure of numerical performance, where ideally no differences are detected. This test setup was initially designed for an advection problem. It is also a suitable test in our case, as we describe the free–surface evolution as an advection problem. This test is performed on a mesh with resolution of $N = 1000$, corresponding to $dx = dy = 0.01$ m. The time step used to solve Eq. B2 is set to $0.000707$ s, which corresponds to a Courant number of

$c_{max} = 0.1$ and a fixed value for the velocity of $10 \text{ ms}^{-1}$. A Dirichlet boundary is set so that $u = u_0$ on the entire domain boundary. As a vertical velocity component is missing and a zero mass balance rate is prescribed, total mass has to remain constant in this test.

### 5.2 Real–world glacier tests

To show the capabilities of the simulation framework for a real–world setting, the temporal evolution of a glacier geometry

based on measurements of South Glacier in southwest Yukon, Canada ($60°49'34"$, $139°07'34"$) is simulated. This valley glacier has an area of $5.3 \text{ km}^2$ and spans an altitudinal range from $1970 - 2960$ m (Flowers et al., 2016).

In order to derive an initial glacier geometry, we use datasets provided in the supplementary of Farinotti et al. (2017), which are based on Wilson et al. (2013) for the surface elevation and the simulation results of Maussion et al. (2019) for the bed geometry. These datasets have a spatial resolution of $20$ m. The computational domain spans a horizontal area of $4.96 \times 6.0$

km, covering the glacier area and its surroundings.

For numerous real–world applications, some kind of preprocessing such as smoothing will be applied to DEM data derived from Lidar or remote sensing measurements. This serves to reduce noise and related potential unrealistic steep surface features and/or to represent only the level of detail that is required for the problem at hand. For this purpose, we smooth the surface elevation data with a Gaussian filter and a standard deviation ($\sigma$) of $2$ pixels equivalent to $40$ m. A surface topography is created

with this smoothed field interpolated to $2$ m spatial resolution using bilinear interpolation. An ice thickness field is computed as the difference between the coarse surface ($S$) and bed ($B$) elevations and the same smoothing and interpolation procedure is applied before subtracting from the final surface elevation field ($S$) to provide a bed topography ($B$). All DEM data processing has been performed with GRASS GIS (GRASS Development Team, 2018).

A satellite image of South glacier and the surface elevation of the smoothed 3D glacier geometry is shown in Fig. 3. In all

figures related to the real–world glacier test cases, the top view panels show the same horizontal extent as shown in Fig. 3.

The mass balance rate representations we apply are chosen to best test the simulation framework performance under realistic conditions. However, the computed test results are by no means glacier evolution predictions under current or projected climatic conditions and only serve the purpose to demonstrate simulation framework capabilities. Two different tests are performed

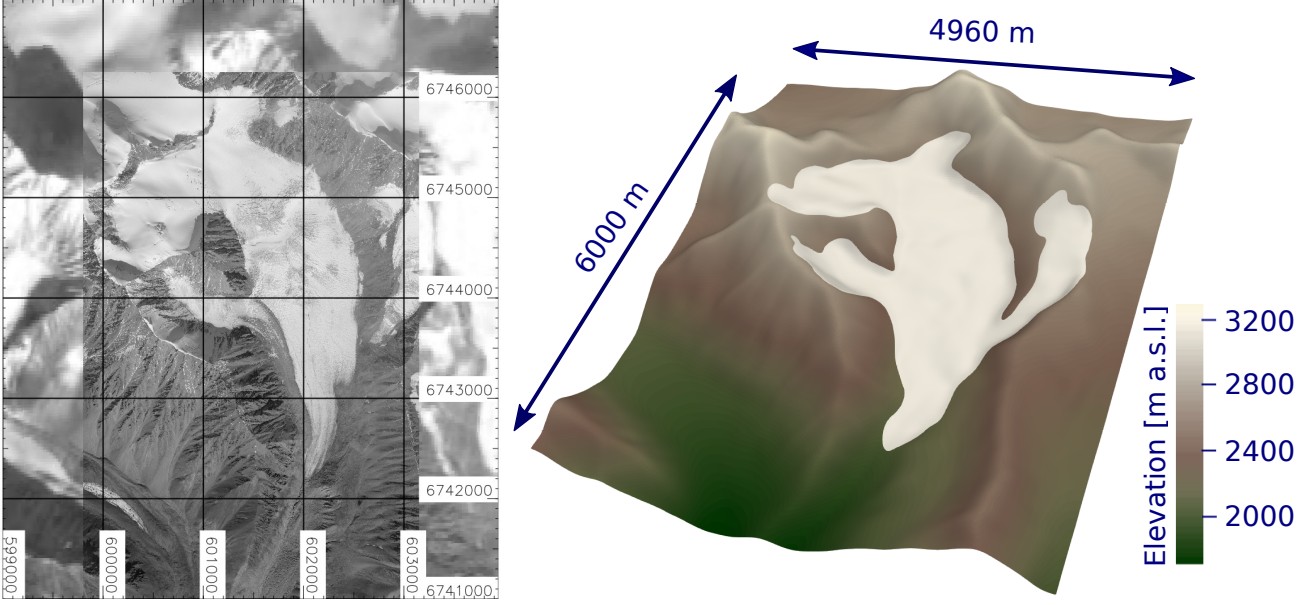

**Figure 3.** Left panel: satellite image (2009) of South Glacier in southwest Yukon, Canada, high resolution image from Schoof et al. (2014) and background filled with Sentinel 2 imagery (2018), courtesy of USGS. The coordinates are given in UTM zone 7 north. Right panel: initial surface elevation of the computational domain. Both panels show exactly the same horizontal extent.

using (a) a zero surface mass balance rate and (b) a synthetic elevation dependent surface mass balance rate. For the velocity computations, the normal velocity component is set to zero at the boundaries, so that the domain boundaries act as walls for ice flow. In case of the free–surface computations, natural (Neumann) boundary conditions are applied. The simulations are performed on unstructured 3D and 2D meshes of approximately $40\,\mathrm{m}$ (Test 1 and Test 2) and $20\,\mathrm{m}$ spatial resolution (Random
5   glacier test).

### 5.2.1 Test 1

The free–surface of the glacier evolves purely due to ice flow. For this purpose, the mass balance rate $\dot{a}$ in Eq. B2 is set to zero, so no mass is added or removed from the system. The surface evolution time step is set to two years, hence new ice velocities are computed every two years.

10  ### 5.2.2 Test 2

The aim of this test is to show the effect of a spatially–distributed, time–evolving mass balance rate. For this purpose, we derive a simple mass balance rate from the mass balance dataset provided in the supplementary of Farinotti et al. (2017) based on Wheler et al. (2014). This mass balance rate is expressed by two piece–wise linear functions representing the accumulation and ablation rate purely as a function of elevation separated at an elevation of $2570\,\mathrm{ma.s.l.}$ The mass balance rate is applied over





the whole domain, causing strong accumulation in the surrounding initially non–glaciated slopes. This is not quite realistic as snow redistribution processes (e.g. avalanches and wind drift) are not simulated in our approach, but appropriate to test the simulation framework capabilities. The resulting mass balance rate field is updated according to changes in surface elevation within each computational time step, whereas a new velocity field is computed every year.

## 5.3 Random glacier test

In this test, the same initial mesh geometry is chosen as in Sect. 5.2.1, but randomly generated input fields for ice velocity and mass balance rate are used that show unnaturally strong, spatially and temporarily varying gradients. In this manner, the simulation framework is subjected to an extremely demanding problem in order to demonstrate its robustness in terms of numerical stability, as the randomly generated input fields also introduce extremely strong gradients in Eq. B2. To generate the input fields, the maximum velocity of each component for the initial glacier geometry is computed, which are: (a) $5.1 \, \mathrm{ma}^{-1}$ in x-direction, (b) $-6.2 \, \mathrm{ma}^{-1}$ in y-direction and $-2.2 \, \mathrm{ma}^{-1}$ in z-direction. From these maximum values, a random noise is subtracted that varies within $100 \, \%$ of the maximum velocity. The mass balance rate is chosen to be significantly negative and set to $-8 \, \mathrm{ma}^{-1}$ and a random noise is subtracted that varies within $200 \, \%$ of this value. A negative mass balance rate causes a decrease of surface elevation and thereby potential contact with the bed constraint. Hence, a negative mass balance regime provides a harder test for the simulation framework. The mass balance rate is updated within every computational time step, whereas velocities are updated every $0.25 \, \mathrm{years}$. The frequent update is necessary in terms of stability, as e.g. there is a strong change in surface elevation due to high values of mass balance rate.

## 6 Results

### 6.1 Benchmark tests

#### 6.1.1 Test A

Fig. 4 displays the initial geometry represented on all four different mesh resolutions $N = 125, 250, 500, 1000$ (corresponding to $dx = dy = 0.08, 0.04, 0.02, 0.01 \, \mathrm{m}$) used in this benchmark. The initial pyramid volume is $V_{\mathrm{init}} = 0.37\dot{6} \, \mathrm{m}^3$.

We evaluate the simulation framework capability to simulate Test A for all four different mesh resolutions by comparing numerical results with an existing exact volume change solution (Eq. 6). In Fig. 5 (upper panel), the volume evolution according to the exact solution (black line) is displayed alongside the four numerical results (coloured lines). Overall, a very good agreement with the exact solution is observed. To produce such an agreement is numerically quite a challenging task as the variational inequality problem (Eqs. 1 and 3) has to be solved correctly over and over again to re–create the exact solution volume decrease. To highlight the subtle differences that remain, Fig. 5 (lower panel), displays the absolute difference between the exact and the respective numerical solutions. As mesh resolution increases, the differences decrease and thus the simulation framework converges towards the exact solution.





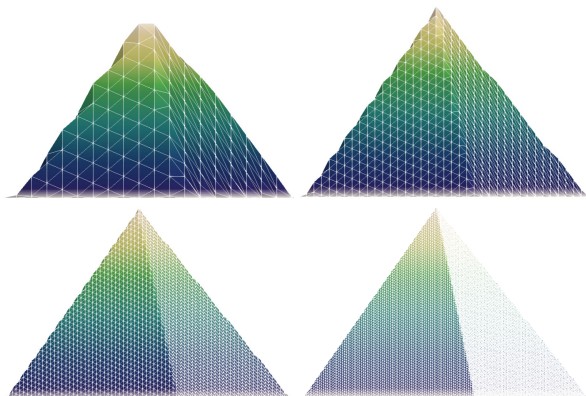

**Figure 4.** Initial pyramid geometry from left to right for mesh resolutions $N = 125, 250, 500, 1000$ to visualize the quality of feature representation on these meshes.

Testing for shape preservation of the pyramid as it gets advected in the prescribed velocity field is carried out in detail in benchmark Test B below. Nevertheless, we compare the pyramid's shape at $t = 4.5\,\mathrm{s}$ in Fig. 6, which also displays the pyramid at times $t = 0, 1.5, 3.0, 4.5$ and $6.0\,\mathrm{s}$ for the mesh resolution $N = 1000$. The sides of the pyramid are very well preserved which is a quality indication for the advection implementation. However, the apex point is not equally well preserved and a smoothing

of the pyramid top can be clearly seen in Fig. 6 (right panel). This smoothing is caused by the continuous function space in which we represent surface elevation and probably some unavoidable numerical diffusion.

Computations of this benchmark test with reduced–space or semi–smooth methods for mesh resolutions $N = 125, 250$ have produced equally good results (not shown here), however we find the semi–smooth based solver to converge slower and less stable. Thus we recommend reduced–space methods for our application.

Overall, benchmark Test A demonstrates clear convergence and strict volume (i.e. mass) conservation for our simulation framework. Both are paramount simulation framework properties that demonstrate an adequate numerical implementation and thus the applicability of the simulation framework for studying free–surface evolution problems.

### 6.1.2 Test B

The results of the swirling flow test are shown in Fig. 7. The simulation framework is capable of reproducing the deformed

shape of the pyramid in every point in time and to recover the edges of the pyramid at the end of the simulation. In the left panel of Fig. 7, a top view of the results at the start and after half of the simulation duration are shown. The arrows indicate the direction of movement throughout the simulation. In the right panel of Fig. 7, the reference shape of the pyramid and numerical solution are compared. In comparison to the reference shape, the pyramid's apex and edges become smoothed, as shown in the inserts, due to the same intrinsic FEM properties as described in Test A above. Mass is conserved up to $99.5\,\%$, however this

performance metric of course strongly depends on the mesh resolution.

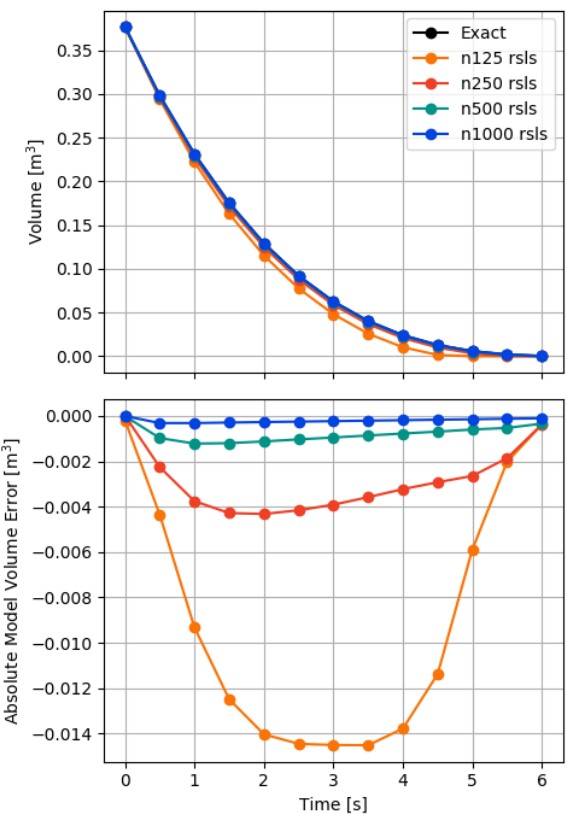

**Figure 5.** Upper panel: Comparison of the exact pyramid volume decrease (black line, Eq. 6) with simulation results (coloured lines) for mesh resolutions $N = 125, 250, 500, 1000$. "rsls" indicates the usage of reduced–space methods to solve the variational inequality. Lower panel shows the absolute difference between the exact solution and simulation results to demonstrate convergence as the mesh resolution increases.

## 6.2 Real–world glacier tests

### 6.2.1 Test 1

In Fig. 8, the evolution of the glacier surface for a period of $100$ years is shown. A decrease in surface elevation in the upper glacier area and its increase in the lower reaches can be observed which illustrates the expected mass transport. Due to the prescribed mass balance rate of $0$ ma$^{-1}$, the expected total mass change is zero. Throughout the entire simulation of $100$ years, the maximum mass change per time step is $0.00032$ % of the total ice mass. The total ice mass at the end of the simulation is $99.988$ % of the initial ice mass.



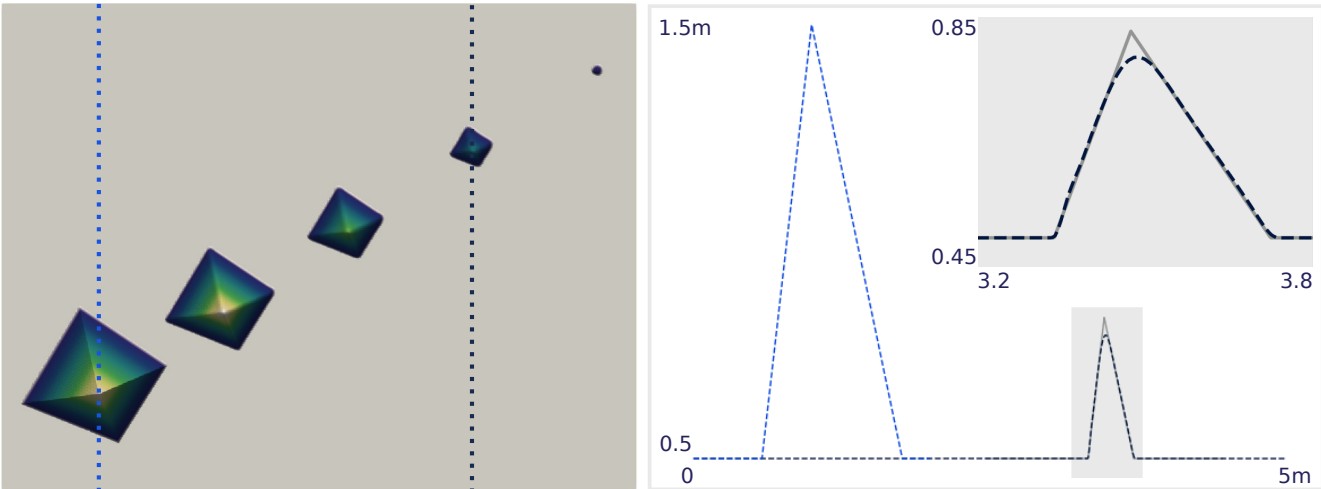

**Figure 6.** Test A: top view of simulation results for $N = 1000$ resolution at times $t = 0, 1.5, 3.0, 4.5, 6.0$ s. Dashed vertical lines indicate the location of the respective profiles in the right panel with the dashed light blue line displaying the profile at time $0$ s. Simulation results are shown by the dashed dark blue line and compared to the analytical solution (grey line) at time $t = 4.5$ s. The small panel on the right show a zoom of the grey rectangle on the main panel.

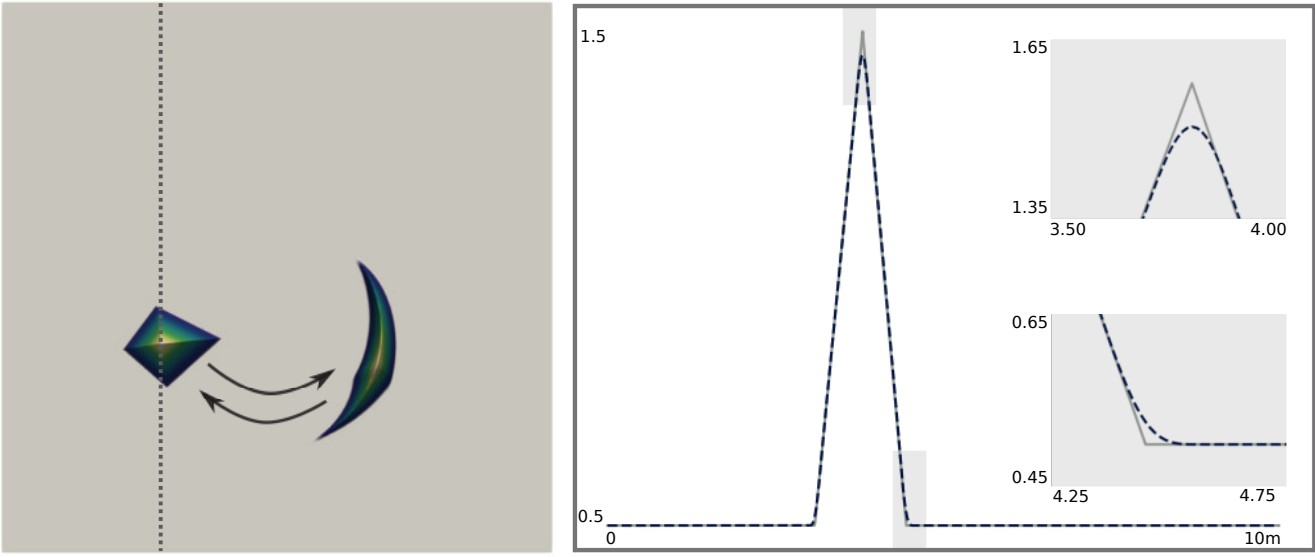

**Figure 7.** Test B: Left panel shows top view of simulation results for the $10$ m x $10$ m domain at time steps $0$ s and $0.75$ s. The dashed vertical line indicates the location of the respective profiles of the simulation results (dashed blue line) and the analytical solution (grey line) at the final time step of $1.5$ s shown in the right panel. The two small inserts show zooms of the grey rectangles on the main panel.





**Figure 8.** Test 1 (zero mass balance rate): top view of difference between initial surface elevation and simulation results at time steps 50 years and 100 years and corresponding profiles with shaded surface topography in brown in the lower panels.

#### 6.2.2 Test 2

In this test, an elevation dependent mass balance rate is included. The simulation results for a period of 100 years are shown in Fig. 9. There are regions of mass gain and mass loss within the computational domain. Below an elevation of 2570 m, the mass balance rate is negative, hence the lower glacier experiences significant melt and decreases in length and height. At elevations

5    greater than 2570 m, accumulation occurs, causing an increase of surface elevation, also beyond the initial glacier margin.





**Figure 9.** Test 2 (measured mass balance rate): top view of difference between initial surface elevation and simulation results at time steps 50 years and 100 years and corresponding profiles with shaded surface topography in brown in the lower panels.

With increasing ice thickness, these newly formed ice masses start to move downslope and develop advancing glacier fronts. If these fronts advance into ice free terrain, at some point the strict solver convergence criteria might not be met anymore when using the Crank–Nicholson time discretization (see Sect. 7 for details on this problem). In the course of this simulations, this happens only for one single time step at 86 years. However, if the total variation diminishing Runge Kutta scheme is used after

5    a time step of 80 years, solver convergence is guaranteed also for this situation.

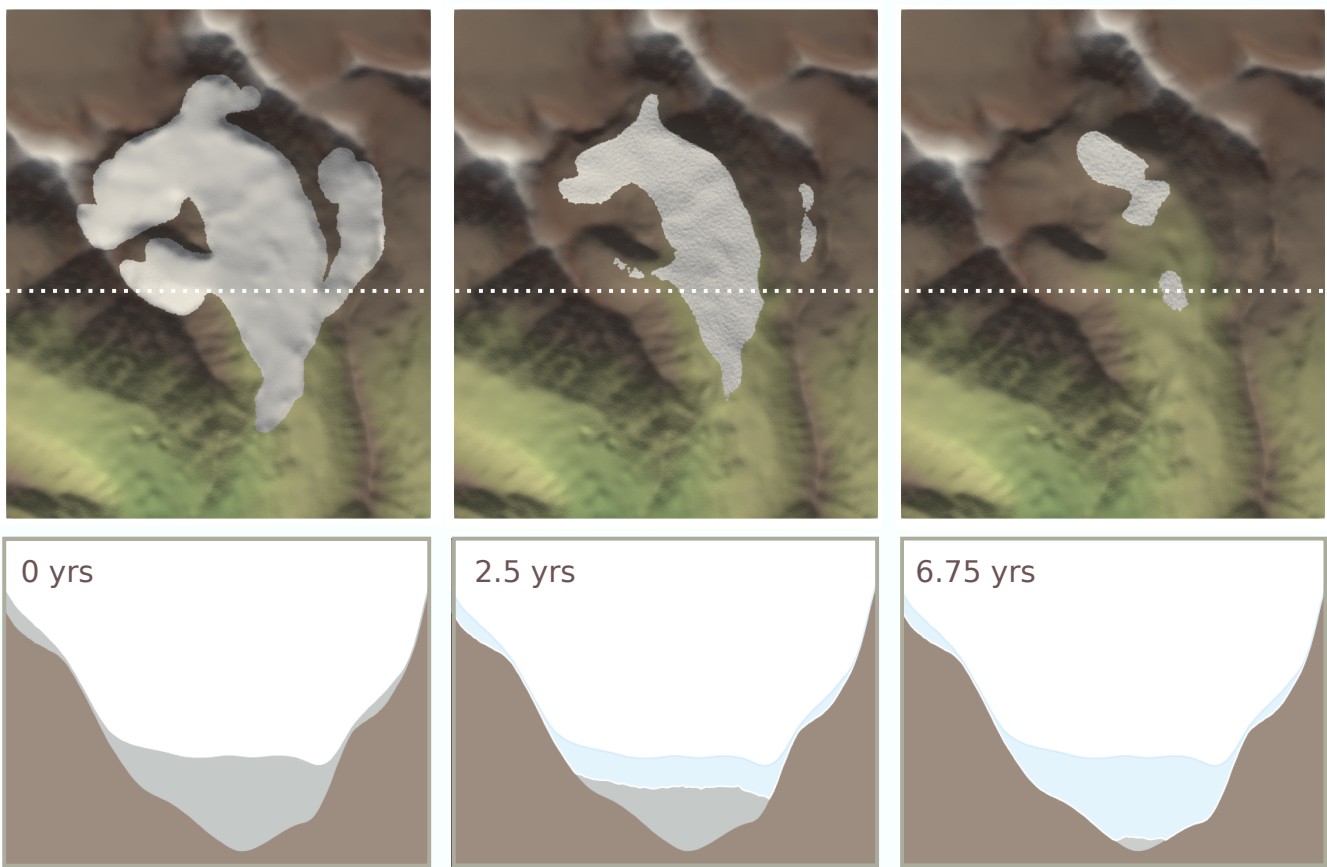

**Figure 10.** Test 3 (random mass balance rate): top view of surface elevation that exceeds the bed constraint in white and bed constraint in colour at the start of the simulation, at 2.5 years and 6.75 years. Corresponding profiles of surface elevation (silver), bed constraint (brown) and initial surface elevation (light blue) in the lower panels. The elevation on the y-axis is shown on a 4:1 scale, thus quite exaggerated.

### 6.3 Random glacier test

In this test, the negative mass balance rate removes the glacier completely after a total of 11.25 years, which is simulated with 225 computational time steps. In Fig. 10, the initial surface elevation and the evolution of the free–surface for two distinct time steps (2.5 and 6.75 years), as well as the corresponding profiles are shown. Due to the strong spatial variations in the prescribed velocity and mass balance fields, the glacier surface becomes heavily undulated. The simulation framework is capable of reproducing this behaviour without introducing any numerical instabilities and thus demonstrates its capability to handle highly complex and variable input fields.



# 7   Stability issues of finite element methods for scalar, convection–dominated equations

When solving the transport problem given in Eq. 3, we have to rely on numerical methods to describe complex glaciological problems (as presented in Sect. 5.2) that have arbitrary initial values and arbitrary source functions (Bueler et al., 2007). We require these numerical approximations to be mass conserving and restricted to an admissible interval as prescribed by

the local constraints (e.g. bedrock elevation). Eq. 3 can be seen as an advection–diffusion equation in the hyperbolic limit, where advection is predominant and hence, it falls into the category of advection–dominated advection–diffusion problems. It is a known problem that for this kind of partial differential equations, standard continuous Galerkin FEMs can lead to the development of unphysical spurious oscillations (e.g. John et al., 2018b). These occur at sharp layers, which is a term referring to locations where strong gradients in the solution are present. The width of sharp layers is typically much smaller than the

mesh size and hence cannot be resolved by these methods (John et al., 2018b).

For this reason, from the outset, we included the SUPG stabilization approach (see Sect. 3). Previous studies have shown that this method efficiently inhibits the development of spurious oscillations studies by introducing numerical diffusion in streamline direction (e.g. Bochev et al., 2004). The SUPG stabilization technique proved to be effective for many of our glacier tests as well, given the solution is sufficiently smooth.

However, in performing tests for a diverse set of mass balance conditions, we occasionally observed unphysical spurious oscillations, if very steep, advancing glacier fronts developed, i.e. the solution is non-smooth for the respective mesh size. Motivated by this observation, we developed an idealized hill test to study these stability issues in detail and analyse the performance of different stabilization schemes for this application. The idealized hill test is explicitly designed to test the robustness of the simulation framework under the most demanding conditions. To study the problem in detail we distinguish

between two cases, steep glacier fronts advance into (i) an already ice–covered domain or (ii) initially ice–free terrain and hence, the bed constraint is potentially affected. In this test, a prescribed elevation–dependent mass balance rate leads to ice accumulation on an Gaussian–shaped hill (see Fig. 11(a)) that is (i) initially ice–free, or (ii) on the same geometry but with an initial ice layer of $10 \, \mathrm{m}$ thickness covering the entire domain. The mesh of the initial geometry is shown in Fig. 11a. For both cases, the mass balance rate is prescribed to increase linearly from for case (i) $0 - 6 \, \mathrm{ma}^{-1}$ (at $9 - 100 \, \mathrm{ma.s.l.}$) and for

case (ii) from $0 - 5 \, \mathrm{ma}^{-1}$ (at $0 - 100 \, \mathrm{ma.s.l.}$). This leads to accumulation of large amounts of ice in a yearly interval causing the build–up of ice which is transported downslope in a kinematic wave. The resulting ice flow causes the ice front to steepen when it reaches the foot of the hill, hence posing a severe test for the simulation framework. The test runs for $30 \, \mathrm{years}$ with a yearly velocity update.

The resulting surface elevation after $24 \, \mathrm{years}$ using the SUPG stabilization technique and Crank–Nicholson time discretiza-

tion is shown in Fig. 11b and a cross–profile of surface and bed elevation is shown in Fig. 11c for configuration (ii). The corresponding results for configuration (i) are shown in Fig. 13, but for $23 \, \mathrm{years}$ of simulation. We chose to show the resulting surface elevation for the simulation time step where oscillations are most prominent, in order to best discuss their characteristics. The spurious oscillations in the solution are clearly visible, however they remain stable for both cases, i.e. they do not increase significantly over simulated time. When ice flows into a previously ice–free region, spurious oscillations develop but




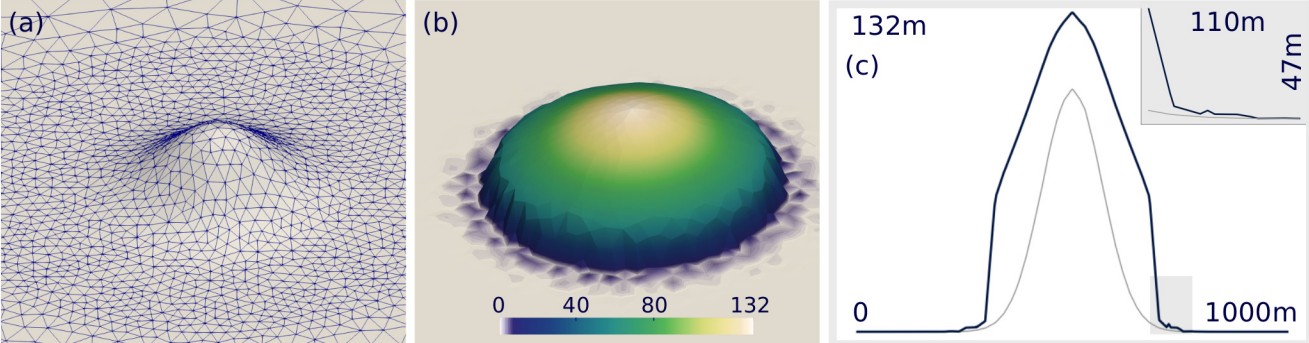

**Figure 11.** In (a) the ice–free geometry mesh formed by a gaussian–shaped hill in flat terrain and in (b) the simulated surface elevation after 20 years (standard continuous Galerkin stabilised with SUPG) is shown with a cross–profile of surface (blue) and bed (grey) elevation in (c).

the negative part of the oscillations most likely becomes efficiently suppressed by the constrained solver. However, if large enough spurious oscillations develop, this eventually can lead to non–convergence of the constrained solver due to the interference with the bed constraint. When the domain is entirely ice–covered, both positive and negative oscillations can arise, but as the bed constraint is not affected, solver–convergence is still achieved even if oscillations develop.

## 7.1 Solution strategies

We have implemented a diverse set of different stabilization techniques in order to find a model setup that is capable of simulating the advance of steep glacier fronts in a numerically stable manner, in particular we aim for simulations that reproduce steep fronts but without spurious oscillations and meeting stringent convergence criteria.

Firstly, in addition to the SUPG method, we tested the following spatial stabilization approaches: (1) continuous interior penalty (CIP, Burman and Hansbo (2004)), (2) Spurious Oscillations at Layer Diminishing (SOLD, John and Knobloch (2008)), (3) 4th–order bubble–enriched functions (BR, e.g. Brezzi et al. (1992)) and (4) discontinuous Galerkin with upwinding (DG). No separate implementation for the Galerkin least–squares stabilisation method (GLS) can be given as GLS becomes equivalent to the SUPG method (e.g. Donea and Huerta, 2003) for hyperbolic equations as in our case. All of these stabilization approaches either add a new term to the variational form of Eq. 3 and/or use a different type of function space. Hence, they can readily be implemented in the model code thanks to the versatility of the FEniCS software. Details of the implementation of individual methods are provided in the Appendix A and the open–source model code accompanying this manuscript. There is another class of spatial stabilization techniques, flux–corrected transport schemes, which showed promising performance in previous studies. However, more analyses for the application in case of anisotropic meshes and non–linear problems are required (e.g. Barrenechea et al., 2018). We did not include a representative method for this class, as these operate directly on the algebraic level, implementation is not as straightforward as for those schemes which modify the variational form of Eq. 3.





Secondly, we introduce adaptive mesh refinement (AMR) in order to increase local mesh size in the vicinity of sharp layers, but without drastically increasing computational costs which would be the case for mesh refinement of the entire domain. The SUPG method is supposed to work more efficiently with increasing mesh size and hence, increasing capability to resolve sharp layers. The adaptive mesh refinement scheme is based on local surface slope and ice thickness. If the surface slope and ice thickness at any point exceed a threshold, an attractor field is introduced in the gmsh file, whereby refinement of the specified region is facilitated. The mesh refinement is performed once per surface evolution time step, likewise for the 3D and 2D mesh.

Thirdly, in addition to the semi–implicit Crank-Nicholson time discretization, we test the Backward–Euler- (BE) and the second–order Runge-Kutta (RK2) method. For the RK2 method, we combine the SUPG and a SOLD method in terms of spatial stabilization, similar to Brinkerhoff and Johnson (2013). The RK2 time discretization increases stability due to the total variation diminishing property as well as by increasing the coupling of free–surface evolution and velocity computations.

Finally, we use a structured mesh (ST) and the SUPG stabilisation approach in order to test the influence of mesh anisotropy.

### 7.1.1 Simulation results

Simulation results for configuration (i) are shown in Fig. 12. Regarding the different spatial stabilization techniques for configuration (i), all of them produce stabilized solutions, meeting the required solver convergence criteria. However the results are display spurious oscillations of varying magnitude, except for the CIP method. As solver convergence is guaranteed, and the oscillations remain stabilized, this is a manageable problem.

However, if ice flows into an ice–free region (configuration(ii) (see Fig. 13 and Fig. 14), results of the SUPG, BE and approaches display oscillations, and do not guarantee solver convergence throughout the simulation period of $30\,\text{years}$. Compared to the SUPG results, the Backward–Euler time discretization leads to a slightly smoother glacier front and the magnitude of the oscillations is smaller. The SOLD and BR method do guarantee solver convergence throughout the entire simulation period, however, the results show spurious oscillations of similar magnitude. Employing the CIP method to stabilize Eq. 3 provides oscillation–free results and stringent solver convergence, however comes at the cost of significant artificial smoothing. The amount of smoothing depends on the choice of the stabilization parameter $\gamma$ (see Eq. A5). Performing a set of tests with different choices of $\gamma$, we determined $\gamma = 0.75$ to be a reasonable choice for this specific test, as greater values lead to drastic smoothing and smaller values lead to enhanced oscillations. The SUPG method, in combination with adaptive mesh refinement leads to a decrease of spurious oscillations, they occur at a later stage and solver convergence is guaranteed throughout the simulation period of $30\,\text{years}$. However, as soon as the advancing fronts become close to vertical also the distinctly finer mesh size is not capable of resolving the extremely sloping surface adequately in a continuous function space.

If the second–order Runge–Kutta time discretization is used, oscillations become smaller and solver convergence is guaranteed throughout the simulation period of $30\,\text{years}$. However, computational costs are drastically increased, as velocity updates are required for each computational time step, and those are by far the most computationally expensive tasks. If simulations are performed on structured meshes, spurious oscillations still develop to some degree, but the SUPG method proves to produce numerically stable results regarding solver convergence. The superior performance in case of structured meshes seems to stem





from the fact that the stabilization parameter $\tau$ (see Sect. 3) is mesh size dependent. The major drawback of using regular meshes is that we lose the ability to employ localized mesh refinement and to represent complex topographies in a detailed manner.

## 7.2 Potential remedies

The presented study suggests that, at the moment, solutions that produce oscillation–free FEM simulations with stringent solver convergence criteria, for the case of steep advancing glacier fronts, based on a time–dependent, non–linear PDE solved on an anisotropic mesh, but without introducing significant artificial smoothing or unaffordable mesh sizes, seem to be still somewhat out of reach. A comprehensive study with the elucidating title: 'Finite Elements for Scalar Convection-Dominated Equations and Incompressible Flow Problems - a Never Ending Story?' by John et al. (2018b) highlights the remaining problems and open questions in this field.

If the solution is sufficiently smooth, the SUPG method is a valid choice for stabilizing the simulations even if the bed constraint is affected. However, if steep advancing fronts develop, further stabilization is favourable. All tested stabilization approaches provide stabilized solutions that meet the stringent solver criteria for configuration (i) whereas not all of the presented stabilization approaches achieve strict solver convergence for configuration (ii) where ice flows into ice–free regions. Stabilized solutions come either at the cost of significant smoothing of sharp layers, drastically increased computational costs or the results are not entirely free of oscillations. For highly detailed studies of glacier geometry change for complex cases, techniques that introduce significant artificial smoothing (as e.g. CIP method or Backward–Euler time discretization) are unfavourable, as e.g. steep glacier fronts become smoothed. However, if smoothing is acceptable for the problem at hand, these approaches can enhance stabilization and therefore the likelihood of convergence. If smoothing is not desirable, (a) a Runge–Kutta time discretization (as suggested in Brinkerhoff and Johnson (2013)) and, or in combination with, (b) high spatial resolution through adaptive mesh refinement or (c) the use of structured meshes proved to be most promising in providing high quality results.

In the case of advancing fronts into ice–free regions, another important remedy is the partition of the domain into subdomains. If Eq. 3 is solved only where ice is flowing, oscillations cannot develop outside of this domain. If the hill test is performed without the partitioning into subdomains, spurious oscillations greatly increase in magnitude and spatial extent and cause solver divergence already at an earlier stage. Hence, reasonable subdomain partitioning actually serves as another remedy for the development of spurious oscillations and also greatly helps to increase solver convergence.

## 8 Discussion

From a mathematical perspective, solving this free boundary problem allows us to provide a physical treatment of a retreating or advancing ice margin. This is not the case in most lower order ice flow models, which often utilize numerical schemes that



include *ad–hoc* treatments for the free ice margin (cf. Jarosch et al., 2013; Bueler, 2016, for an overview). Our approach eliminates the need to perform any *ad–hoc* post–processing step dealing with negative ice thickness resulting from regions where the negative mass balance rate per time step is greater than the actual thickness of remaining ice. It has been demonstrated that such post–processing steps often result in mass conservation violations (e.g. Jarosch et al., 2013). As mass conservation

is not inherent for FEM methods, we chose to thoroughly check mass conservation for different cases where the actual mass change is known. In benchmark test A, a non–zero mass balance rate is applied but we can compute the expected mass of the pyramid at any point in time using its reference shape, whereas no mass change is expected for Test B and glacier Test 1. When simulating the evolution of a glacier including a realistic representation of glacier mass balance rate, mass is constantly added or removed from the system. This is a non–trivial problem implying that overall mass can change within the course of the

free–surface evolution computations. Furthermore, expected mass changes cannot be computed directly without any *ad–hoc* procedures, due to the constraint on the surface elevation. To ensure mass conservation for FEM methods, mesh resolution is crucial but it is hard to estimate the required mesh size *a priori*. Convergence tests can indicate the required mesh size to ensure mass conservation and correct representation of glacier geometry changes due to ice flow and mass balance rate. Furthermore, one has to keep in mind that the velocity field itself is an approximation as it is solved via FEM on a mesh of finite spatial

resolution. Hence, any errors in this quantity could propagate into the free–surface evolution. For example, in a recent study by John et al. (2018a), it has been shown that for a mantle convection model, a coupled system of the stationary incompressible Stokes equations and the convection-diffusion equation, the error of the solution in one problem has an impact on the solution of the other problem.

The simulation framework presented here performs simulations on meshes that also include the surrounding landscape of a glacier. This allows us to represent processes that require knowledge about the surrounding topography, and to simulate glacier advance into previously ice–free terrain. Furthermore, as the computational domain can be partitioned into different subdomains it is possible to solve individual forms of Eq. 3 for each subdomain all at once. Hence, individual processes can be included for each subdomain, and this can reduce computational costs, for example, if computationally expensive tasks are only

solved on relevant parts of the domain. The definition of these subdomains is straight forward and can be updated whenever required, as long as the delineation is based on ice thickness, surface slope, velocity or mass balance rate fields. Other criteria could be implemented if needed. In the presented simulations, the domain is partitioned according to glacierized and ice–free regions, so that for ice–free regions, the free–surface evolution can be computed solely as a function of the mass balance rate and potential effects of the artificial ice layer can be excluded. Furthermore, as shown in Sect. 7.2, this is also beneficial in

terms of numerical stability.

We have tested the simulation framework capabilities for a diverse set of problems ranging from specific benchmark tests, to complex applications for a real–world glacier geometry. The results of the benchmark tests show that the simulation framework is able to reproduce analytical results for challenging geometries (Test A) and complex flow cases (Test B) and it conserves





mass well even in the case of overall mass change (Test A). However, a thorough stability analysis showed that if very steep, advancing glacier fronts develop, spurious oscillations might arise.

The SUPG approach as well as the other stabilization schemes presented in Sect. 7.1, could successfully stabilize the solution, but had variable success in suppressing spurious oscillations at sharp layers. In the case of glacier advance into ice–free terrain,

these oscillations affect the bed constraint and hence can eventually lead to convergence problems of the constrained solver. We have identified that dividing the entire domain into subdomains, and solving the full form of Eq. 3 only in the relevant domains, serves as a potential remedy for this problem. If the full form of Eq. 3 is solved in a subdomain defined by the glacierized part including a velocity dependent bufferzone but not reaching beyond this zone, spurious oscillations can successfully be inhibited by restricting the advection term to these parts of the domain.

Also, the weak coupling between velocity computations and free–surface evolution (as a consequence of linearization for the Crank–Nicholson and Backward Euler schemes (see Sect.)) is not favourable if steep, advancing glacier fronts develop. In this case, a total variation diminishing time discretization, as e.g. a Runge–Kutta scheme, performs much better in terms of numerical stability and accurate representation of surface elevation, as has been proposed in Brinkerhoff and Johnson (2013). In Test 2 of the glacier test, the Crank–Nicholson scheme fails in terms of solver convergence when steep glacier fronts ad-

vance into ice–free terrain for one time step. Using the Runge–Kutta method and additional SOLD stabilization, provides stable results and guarantees solver convergence also for this situation. This example illustrates that appropriate methods have to be chosen according to the problem at hand, as e.g. for complex situations like steep advancing fronts, a tighter coupling between velocity computations and free–surface evolution is favourable. However, using the Runge–Kutta time discretization comes at the cost of drastically increased computational effort as the number of required 3D velocity computations increases.

In the case of smooth solutions, the Crank–Nicholson and Backward Euler schemes, i.e. a much weaker coupling of velocity computations and free–surface evolution, proved to be an acceptable compromise between low computational costs and accuracy. Furthermore, the use of structured meshes or adaptive mesh refinement proved to also decrease the magnitude of spurious oscillations and accordingly the likelihood of non–convergence (see Sect. 7.1). This stems from the fact that (i) the SUPG stabilization parameter is mesh size dependent and structured meshes are favourable and that (ii) the capability to resolve sharp

layers increases with increasing spatial resolution, which is the case for adaptive mesh refinement. In case of non–convergence or if numerical instabilities develop, we suggest to check the time stepping and time discretization scheme, mesh quality and mesh size and to revisit the definition of subdomains. In case of coarse meshes, the subdomains can reach far into the artificial ice layer which is not favourable and facilitates the development of oscillations. This means that for complex geometries, a minimal spatial resolution has to be provided depending on the complexity, e.g. steepness, of the actual topography, which is

also crucial in terms of mass conservation.

Simulations usually spend most of the overall computation time ($> 95\%$) at the Stokes problem. Due to the complex mesh modifications and direct operations on the degrees of freedom, the free–surface computations are not parallelized, which is not problematic regarding their share of the overall computational costs.






## 9    Conclusions

The free–surface evolution of flowing ice formulated as a variational inequality (proposed earlier by Jouvet and Bueler (e.g. 2012)) has been implemented in a new, open–source numerical simulation framework.

This simulation framework automates (1) mesh generation, (2) computation of 3D ice velocity based on a full–Stokes approach, (3) simulation of free–surface evolution due to ice flow and mass balance rate including the glacier surroundings and fully accounting for the constraint of $S \geq \tilde{B}$, as well as (4), generation of updated 3D geometry meshes.

It offers the option to chose different time discretizations and spatial stabilisation approaches as well as adaptive mesh refinement. This is of special interest for detailed studies of glacier geometry change or given complex topographical conditions. These features make it a highly versatile tool for different glaciological applications. The simulation framework can also be applied for highly complex cases of multiple glaciers (including their initial formation) within one domain and for a wide range of glacier mass balance conditions. The option for domain–partitioning, makes it straight–forward to add processes that are relevant for individual subdomains only.

We have proposed a set of benchmarks (c.f. Sects. 5.1 and 7) to evaluate numerical implementations of free–surface flows as variational inequalities. Performing these tests, numerical stability, mass conservation, shape preservation and hence, the ability to reproduce realistic glacier states can be assessed. Our simulation framework performs well in those challenging tests and demonstrates mass conservation even in the case of overall strong mass changes (i.e. mass balance) for sufficiently fine meshes. It is important to test these aspects, as simulation results can look realistic but this does not alone ensure correctness of the results. Furthermore, the idealized hill test simulations highlight potential limitations of the presented approach and provide information which conditions require caution in order to ensure high quality results. To demonstrate the suitability of our approach for real–world glaciological applications, we have presented several results with glaciological input data and simulation periods of 100 years.

The physically consistent, free–surface simulation capability of our framework presented here is not only suitable for glaciological applications, but for many free–surface flows found in geoscience. Our proposed benchmark tests are highly suitable to evaluate numerical implementations of free–surface flows on a general level. Our review of spatial stabilization methods and time discretization approaches will be useful to choose a suitable combinations for a given simulation task as it is apparent that there is no single suitable approach for all glaciological applications. Therefore we hope that our provided benchmarks find wider use in assessing the numerical performance of existing and upcoming models.

*Code availability.* The simulation framework code is available at: http://doi.org/10.5281/zenodo.3734021 and a repository containing the model code and test examples can be found at: https://github.com/awirbel/evolve_glacier. The code is provided under the GNU General Public License v3.0. We suggest to run the model using the singularity container (including an installation of FEnICS v2016.2 and gmsh) provided in the assets of the model code release. Details on how to run the simulation framework and additional information on how to create initial 3D meshes is provided in the Readme file accompanying the model code and in the github repository.





## Appendix A: Stabilized variational forms

We briefly show the variational forms we employed to implement the different stabilization approaches described in Sect. 7.2.

In the following, $\|..\|$ denotes the norm, $[..]$ the jump and $\{..\}$ the average of a quantity. $v$ denotes the finite element test function, $d\Omega$, $ds$ and $dS$ represent integration over cells, exterior or interior facets, respectively. If not stated differently, all

utilized parameters refer to the same quantities as in the manuscript.

The listed stabilization techniques are based on the following variational form of Eq. 3:

$$F1 = \int v\,(S - S_0)\,d\Omega + \Delta t\,(\int v\,u_h \cdot \nabla S_{mid}\,d\Omega - \int u_z\,v\,d\Omega - \int \dot{a}\,v\,d\Omega) \tag{A1}$$

It is important to mention that the described spatial stabilization approaches strongly depend on the choice of the respective stabilization parameters. As it is a field of research on its own how to determine the best choice of parameters, we went with

standard values or suggestions from the literature. Certainly, different choices of these parameters could lead to differences in the results.

### A1 SUPG method (Hughes and Brooks, 1982)

All Functions are defined on a continuous piecewise linear function space. The SUPG method is based on adding artificial diffusion in streamline direction based on the residual $r$ of Eq. 3 and a mesh-size dependent stabilization parameter $\tau$. This

gives the following modified variational form:

$$r = S - S_0 + \Delta t(u_h \cdot S_{mid} - u_z - \dot{a}) \tag{A2}$$

$$\tau = \frac{h}{2\,\|u_h\|} \tag{A3}$$

$$F = F1 + \tau \int u_h \cdot \nabla v\,r\,d\Omega. \tag{A4}$$

### A2 CIP method: Burman and Hansbo (2004)

All Functions are defined on a continuous piecewise linear function space. The CIP method stabilises the gradient jump across element boundaries by adding the following term to the variational form of Eq. 3 including the stabilization parameter $\gamma$:

$$j = 0.5\gamma h^2\,[\nabla S_{mid}, n] \cdot [\nabla v, n]\,dS \tag{A5}$$

$$F = F1 + j. \tag{A6}$$

### A3 SOLD method: modified Burman (John and Knobloch, 2008; Burman and Ern, 2002)

All Functions are defined on a continuous piecewise linear function space. The presented SOLD method adds an additional term to the variation form stabilised with the SUPG method. This second stabilisation term is based on the residual, the SUPG




stabilisation parameter $\tau$, norms of velocity and velocity gradient as well as the norm of the solution gradient.

$$C = \frac{\tau \, \|u_h\| \, r}{\|\nabla S_{mid}\|} \, \frac{\|u_h\|\|\nabla u_h\|}{\|u_h\|\|\nabla u_h\| + \|r\|} \tag{A7}$$

$$F = F1 + \tau \int u_h \cdot \nabla v \, r \, d\Omega + C \int \nabla v \cdot \nabla S_{mid} \, d\Omega \tag{A8}$$

### A4  BR method e.g. Brezzi et al. (1992)

All Functions are defined on a continuous piecewise linear function space, but the solution function, test and trial functions as well as boundary conditions are defined in a 4th order bubble enriched function space. The variational form given in Eq. A1 remains unchanged.

### A5  DG method

All Functions are defined on a continuous piecewise linear function space, but the solution function, test and trial functions
as well as boundary conditions are defined on a discontinuous function space. For stabilization we use upwinding, following the dg-advection-diffusion demo provided within the FEniCS software. We set $\kappa$ to zero, which results in a pure advection equation.

### Appendix B: Time discretization schemes

### B1  Crank–Nicholson scheme

The employed Crank-Nicholson time discretization is of the form:

$$\frac{S^{t+1} - S^t}{\Delta t} = -u_h^t \cdot \nabla_h S^{t_{mid}} + u_z^t + \dot{a}^t, \tag{B1}$$

where $S^{t_{mid}} = 0.5(S^{t+1} - S^t)$ is the midpoint solution and the index $t$ describes time. Here, linearization is applied by using $u_h$, $u_z$ and $\dot{a}$ from the previous time step instead of considering their actual non–linear dependence.

### B2  Backward–EUler scheme

The employed Backward–Euler time discretization is of the form:

$$\frac{S^{t+1} - S^t}{\Delta t} = -u_h^t \cdot \nabla_h S^{t+1} + u_z^t + \dot{a}^t. \tag{B2}$$

We perform linearization by using $u_h$, $u_z$ and $\dot{a}$ from the previous time step.





**Appendix C: Runge-Kutta scheme (Gottlieb and Shu, 1998)**

We employ an explicit trapezoidal method to provide a second-order Runge-Kutta time discretization, following:

$$\tilde{S} = S_t + \Delta t f(t_t, S_t) \tag{C1}$$

$$S_{t+1} = S_t + \frac{\Delta t}{2}(f(t_{t+1}, \tilde{S}) + f(t_t, S_t)) \tag{C2}$$

5   where $t + 1$ is the new time step and $t$ is the previous time step.

*Author contributions.* AW developed the model code with methodological input from AHJ. AW and AHJ designed the benchmark tests. Simulations were carried out by AW whereas AHJ provided computational resources for the more extensive simulations. AW prepared the manuscript with support from AHJ.

*Acknowledgements.* This work was funded by the Austrian Science Fund (FWF), project P28521. Special thanks to Lindsey Nicholson for
10   valuable discussions of the work performed and commenting on the manuscript as well as to Axel Kreuter for invaluable figure presentation recommendations.



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



**Figure 12.** The left panels show results of hill test for stabilized simulations after 24 years using BR (bubble enriched), CIP (continuous interior penalty), DG (discontinuous Galerkin), AMR (adaptive mesh refinement), SOLD (oscillations at sharp layer diminishing) methods, RK2 (Runge–Kutta time discretization), BE (Backward-Euler time discretization) and a ST (structured mesh). The right panels show a cross-profile along the y-axis of surface (blue) and bed (grey) elevation; inlets show a zoom of the opaque grey regions, grey bold line indicates the reference solution using the SUPG and Crank-Nicholson setup (see also Fig. 11).

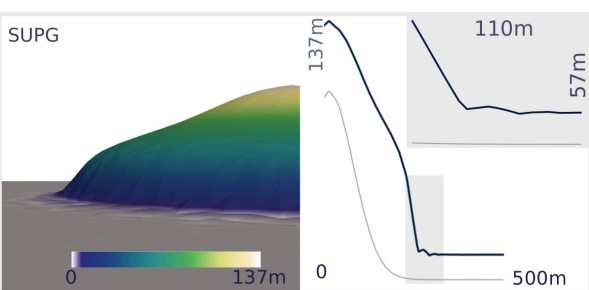

**Figure 13.** The left panel shows the result of the hill test for stabilized simulations after 23 years using the SUPG approach and Crank–Nichsolson time discretization. The right panel shows a cross-profile along the y-axis of surface (blue) and bed (grey) elevation; The inlet shows a zoom of the opaque grey regions.





**Figure 14.** The left panels show results of hill test for stabilized simulations after 23 years using BR (bubble enriched), CIP (continuous interior penalty), DG (discontinuous Galerkin), AMR (adaptive mesh refinement), SUPG, SOLD (oscillations at sharp layer diminishing) methods, RK2 (Runge–Kutta time discretization), BE (Backward-Euler time discretization) and a ST (structured mesh). The right panels show a cross-profile along the y-axis of surface (blue) and bed (grey) elevation; inlets show a zoom of the opaque grey regions, grey bold line indicates the reference solution using the SUPG and Crank-Nicholson setup (see also Fig. 11).