# Peer review of "Inequality constrained Free–surface evolution in a full-Stokes ice flow model (*evolve\_glacier v1.1*)"

_Geoscientific Model Development, 2020_

## Referee Comment (RC1) · Anonymous Referee #1 · 18 May 2020

review of "Free–Surface Flow as a Variational Inequality (evolve_glacier v1.1): Numerical Aspects of a Glaciological Application" by Anna Wirbel and Alexander Helmut Jarosch

This paper presents interesting insights for the solution of the kinematics equation of the upper stress-free surface of a glacier under the constraint that the ice thickness cannot be a negative value. This result in a variational inequality that is solved using the finite element method. The paper test different stabilization schemes to solve this equation and also different time step discretization schemes. Different benchmarks are presented to evaluate the different discretization schemes and validate the implemen-

tation of the method. My overall feeling about the paper is that it deals in a rigorous way to an important numerical problem in glaciology that is not well addressed by many models such that this contribution will help de the community improving their models. But, if I am quite convinced that the material presented here is rigorous and of good quality, the presentation itself of all this material is a bit confusing. At the end, I am not quite sure of how the constraint S > B is enforced. The problem is that the central topics of the paper (solving the free surface equation as a variational inequality) is hidden by a number of technical details (e.g remeshing). For example, in 4.2 which is about how the free surface is solved, it is just said that it is done by solving Eq. (3) and then the remaining part of the paragraph is regarding technical aspects. But how is solved this equation with the constraint S>B is never clearly stated. I would really suggest to focus on the main text on the numerical methods used to solve the free surface equation with the constraint S>B (reduced-space method?) and to separate it from the technical implementation of this equation under a more general model frameworks (solving Stokes, dealing with 2D and 3D meshes, allowing remeshing, etc...). The part 2 should therefore be extended (Mathematical formulation and numerical implementation) such that the variational formulation is given as well as the way the constraint S>B is enforced.

I have some more specific comments below:

- abstract: acronym PETS and SNES should be given (it seems that the explanation of how it is solved is here?) but better would be to explain with words which methods is used to solve it (which tools/modules is used could be given later in the text). In general the abstract should be more focussed on the main points of the paper and avoid the technical aspects (is the fact that there is subdomains, a feature quite classical in FEM, relevant in the abstract?). Moreover, the term/verb "partition" has a special meaning in FEM (parallel computing with partitioned mesh). Here you are referencing to different bodies on which you are solving different equations?

- in the introduction, you should clearly specify that the main topics is the free surface equation under the constraint S>B, but that you eventually also need a Stokes modules

to compute the velocity entering this equation as well as a module to evolve the mesh.

- eq. (2): the effective viscosity is not given. Anyway, is this equation really needed as it is a boundary condition for the Stokes solver, not the free surface?

- line 23, page 4: Crank–Nicholson- -> Crank–Nicholson

- part 3: I found part 3 a bit confusing. For example, page 4, lines 24-30, what does "an update" means? Are you remeshing or just moving the nodes (which ones? only vertically?) such that the surface nodes stay on the surface? And what do you mean by "significant"? How often the mesh is updated and how should be clearly stated. It seems that you are deforming your mesh such that the surface nodes stay on the surface but then remeshing when the displacements of these nodes is too large, but not sure. To my point, to get the correct solution, one has to perform an update of the mesh at each time the free surface variable S has been modified. Not really sure if you are doing that when reading the sentence lines 27-29.

- page 6, line 21: I would suggest to use nodes instead of gridpoints.

- page 7, ine 3: are the Stokes equations also solved in subdomain 2? Anyway, velocity computed on domain 2 should be $\sim 0$ so that Eq. (3) should reduce to (4) even without the subdomain strategy?

- page 7, line 6: can you explain what you mean by a "velocity-dependent" buffer zone"?

- page 7, lines 8-13: an example of where you explain how you solve the free-surface equation under the constraint S>B but just giving the technical details of the modules used, not really the method. All these materials all along the manuscript should be grouped in part 2.

- part 4.3: if you are using a vertically extruded mesh or a completely unstructured mesh should be mentioned here. Also, the last sentence is not very clear (what is the difference between creating a new mesh and remeshing?). A "msh" file is a bit technical. My understanding is that the STL surface has been modified, then the "msh"

file is necessarily modified and therefore a new mesh has to be created. So this should be done each time the variable S has been modified? Also, how the previous solution (velocity, but also S) is interpolated on the new mesh should be discussed.

- page 9, line 6: It seems that S=S_0 is more an initial condition than a Dirichlet boundary condition? It is only set at t=0 not for any t?

- page 9, around equation (5): I would suggest to avoid mixing of _0 and _init? S_0, A_0 and V_0 would be more consistent, meaning the value at t=0. Also, in (5) h should write h_0, but to avoid new variable to be introduced h+z(t) could be replaced by S(t)=S_0 + (u_z + a) t?

- page 11, lines 1-5: what is the boundary condition at the bedrock for the Stokes?

- page 11, line 11: give the SMB that you have used such that the test can be reproduced by other users. In general, the exhaustive setups of these tests that aim to serve here as benchmarks should be given.

- page 12, line 6: 5.2.1 -> 5.2?

- page 12, line 9: specify that Eq. (B2) is given in Appendix 2.

- page 12? line 15: give the value for the computational time step. It seems from this sentence that the Stokes and the free-surface solver are not solved at each time step. This should be explained before in the manuscript. Regarding the advancing and retreat cases, I would suggest to discuss the two situations already in the introduction. At the end, you are presenting both situations which have their own difficulties in terms of numerics (constraint S>B when retreating, front oscillation when advancing).

- page 12, line 22: how much the initial volume of the pyramid is a function of the mesh resolution? There is a dot over 6 in 0.376?

- Figure 5: relative difference instead of absolute would be more pertinent as nevertheless the choice of these values has no real meaning?

- Figure 6 (and others): I would suggest to avoid the grey background and also the frame around panels. I would also suggest to use letter (a, b, c...) to label the different panels of a given figure. "N = 1000 resolution" looks strange (for the resolution N=1000).

- Figure 8: ice thickness would be more informative, or at least the initial S should be drawn on these two panels giving S?

- page 18, line 3: and at other places. Why computational time step as it has a physical meaning. So time step alone would be better.

- page 19, line 28: again, are the velocity and free surface solved using different time steps?

- page 21, line 16: are display?

- page 21, line 18: region (configuration(ii) (see Fig. 13 and Fig. 14), results -> region (configuration (ii), see Fig. 13 and Fig. 14), results

- part 7.2 misses an analysis if the oscillations are created from the free surface equation and/or the Stokes velocity solution in line of what has been shown in John et al. (2018a)? With a perfect velocity field, would one get surface oscillations? With a perfect free surface solver, would one get oscillations?

- Eq. (A1): define $S_0$, $S\_mid$ ($S\_mid$ defined after B1 should be defined here)

- Eq. (A3): what is h here (I guess not the same as in (5))? $h_K$ was used somewhere in the text.

- make title A1, A2, etc... similar

- page 28, line 5: t, t+1 should be defined before

---

## Referee Comment (RC2) · Anonymous Referee #2 · 8 Jun 2020

I enjoyed reading this manuscript and feel it is an importance piece of scientific work. I however agree with the other reviewer that the title is arguably somewhat inexact, if not even a bit misleading. The manuscript does not focus exclusively on the issue of free-surface flow with a pos. thickness constraint, but is in fact much wider in its scope. The discussion about the variation inequality is short and it is actually unclear how it is applied or solved. It is also unclear to what degree this work uses the cited work of Jouver and Bueler, 2012. It appears that the thickness constraint is imply plugged into the PETSc solver. Furthermore, I would have liked to see how the active set of the KKT system is actually updated and when it is considered to have converged. That is to say, if that is indeed the solution method applied. This, and other technical details

are somewhat missing in the manuscript.

I suggest refocusing this work and presenting it more as a new full Stokes ice-flow model. The paper is quite descriptive at times. I would have liked to see the equations and I guess some of the description could be shorted significantly by just listing the equations in their weak form and specifying the FE spaces. I felt the discussion all they way done to page 7 was very much describing the standard approach. Having said that, in the particular case of this journal, this is presumably justified, but I still feel having all the equations listed in one place as a clearly defined mathematical system might be a good option.

As far as I could see, most of the test presented related to how accurately the (SUPG stabilized) mass-transport equation is solved. The test are useful and doing these and similar test is an essential part of the model-development phase. I did not see that the thickness constraint is mass conserving, and the discussion on page 23 suggested that it is in fact not. However, almost all of the tests and the associated discussions revolved around the stabilisation of the surface elevation equation and this mass-conserving aspect was not really addressed or analysed. I in fact doubt that the thickness constraint can be locally mass conserving for any finite time step. I never saw the details of the method, but if this is solved using PETSc as a constraint minimisation problem, then I suspect the corresponding Lagrange multipliers can be thought of as fictitious mass sources.

Equations A2-A4 are referred to as being on variational form, but I do not see a variational form there. Also, should \tau not be inside the integral as it is element dependent and therefore spatially variable? Most of the discussion in the appendixes is presumably 'common knowledge'. I would list these equations as a part of the whole system, but is there any reason to have an appendices on Crank-Nicholson, Runge-Kutta and Backward Euler in a professional journal?

All in all, this is a good manuscript. I know that some of my above comments might

be a bit on the negative side, but then again reviewers are support to help improving things my pointing at things that can/should be improved.
* * *

---

## Author Comment (AC1) · 1 Jul 2020

Anna Wirbel[1] and Alexander Helmut Jarosch[2]

[1]Department of Atmospheric and Cryospheric Sciences, University of Innsbruck, Innsbruck, Austria
[2]ThetaFrame Solutions, Hörfarterstrasse 14, Kufstein, Austria.
**Correspondence:** Anna Wirbel (Anna.Wirbel@uibk.ac.at)

**1   General Response**

We thank both Anonymous Referees for their helpful and constructive comments on our manuscript. As both referees suggested to include more information on the variational inequality formulation, how it is solved and to adjust the structure and focus of the paper and title, we first reply to these general comments that both referees mentioned in this General Response
5   Section.

Regarding the manuscript title, we adjusted it to be more representative of the fact that with this manuscript, we introduce a new free–surface evolution model for ice flow that directly accounts for the inequality constraint. It now reads: 'Inequality constrained Free–Surface flow in a full-Stokes ice flow model (evolve_glacier_v1.1)'

10   Following the general comments on including more information on how the inequality constraint is accounted for, the variational inequality formulation itself, how it is solved numerically, on refocusing the work and it being quite descriptive, we considerably restructured the manuscript. In particular, we added the full variational form, including the respective function spaces and a representation of the spatial stabilisation approaches as well as a description of how incorporating the inequality constraint turns our problem into a variational inequality / non-linear complementarity problem. All of these points have been

15   added in the Section Mathematical Formulation and as the Equations are now all listed within this Section, they are not listed in the Appendix anymore. Regarding the comment on how the variational inequality is solved, we added more details on the numerical method applied ('reduced space method', Benson and Munson (2006)) and on the software (PETSc's SNES solver, Balay et al. (2018a, b, 1997)) that we employ for this purpose, in the renamed Section Numerical Implementation. All the details on preprocessing, mesh generation, remeshing and how to derive the parameters given on the 3D glacier geometry onto

20   the 2D mesh required for the free–surface evolution computations, have now been merged into one Section. Following the referee's comment about the paper being quite descriptive at times, we decided to move this entire Section to an Appendix. In this manner, the technical details about mesh-related issues are not included in the main text, but all the relevant information can still be found within the manuscript. As a description of the variational inequality formulation and details on how this

is solved numerically, as well as on the software packages that are used for this purpose, have been added in the Sections Mathematical Formulation and Numerical Implementation and details on how the parameters are derived on the 2D mesh from the 3D mesh have been moved into an Appendix, the former Section 4.2 Solving Free–Surface Evolution is not required anymore, as more comprehensive information on these issues is now provided in the respective precedent Sections. In the Section Model Chain, we added information on the sequence of velocity computations, free–surface evolution and remeshing for all three time discretization options and now included a flow chart that further illustrates this sequence.

With this substantial changes in the manuscript structure, we hope to have implemented the referees suggestion to improve the manuscript. In the following Sections, we will address the remaining specific comments of each referee point-by-point.

**2 Response to Anonymous Referee #1**

**2.1 Specific Comments**

We also thank Anonymous Referee #1 for all the valuable specific comments and helpful text edits.

**Comment:** *abstract: acronym PETS and SNES should be given (it seems that the explanation of how it is solved is here?) but better would be to explain with words which methods is used to solve it (which tools/modules is used could be given later in the text). In general, the abstract should be more focussed on the main points of the paper and avoid the technical aspects (is the fact that there is subdomains, a feature quite classical in FEM,relevant in the abstract?). Moreover, the term/verb "partition" has a special meaning inFEM (parallel computing with partitioned mesh). Here you are referencing to different bodies on which you are solving different equations?*

**Response:** We added information on the numerical method used to solve the problem as well as the longform of the acronyms, keeping information on the software packages used to for the solving procedure. We do not use the term 'partition' anymore throughout the entire manuscript when we refer to dividing the computational domain into different subdomains for performing different computations, in order to clearly distinguish this from the meaning of partitioning a domain in terms of parallel computing. However, we would suggest to keep a note on the simulation framework's capabilities of dividing the computational domain, to allow computation of different forms of the governing equations all at once.

**Comment:** *in the introduction, you should clearly specify that the main topics is the free surface equation under the constraint S>B, but that you eventually also need a Stokes module to compute the velocity entering this equation as well as a module to evolve the mesh.*

**Response:** In order to highlight the importance of the inequality constraint, we added: 'If the free–surface flow of ice is defined as a variational inequality, the constraint imposed on the free–surface by the bedrock topography, is incorporated directly, thus sparing the need for ad-hoc post-processing of the free boundary to enforce no-negativity of ice thickness (Jouvet and Bueler, 2012; Bueler, 2016a). ' in the Introduction.

**Comment:** *eq. (2): the effective viscosity is not given. Anyway, is this equation really needed as it is a boundary condition for the Stokes solver, not the free surface?*

**Response:** Thanks for this comment, we now describe the effective viscosity $\eta$ and cite its origin. However we choose to keep the Stokes solver boundary condition in the manuscript to highlight the link between the Stokes solver and the free surface flow.

**Comment:** *line 23, page 4: Crank–Nicholson- -> Crank–Nicholson*

**Response:** Done.

**Comment:** *part 3: I found part 3 a bit confusing. For example, page 4, lines 24-30, what does"an update" means? Are you remeshing or just moving the nodes (which ones? only vertically?) such that the surface nodes stay on the surface? And what do you mean by "significant"? How often the mesh is updated and how should be clearly stated. It seems that you are deforming your mesh such that the surface nodes stay on the surface but then remeshing when the displacements of these nodes is too large, but not sure. To my point, to get the correct solution, one has to perform an update of the mesh at each time the free surface variable S has been modified. Not really sure if you are doing that when reading the sentence lines 27-29.*

**Response:** Thanks for this comment. We fully agree, it is important to perform remeshing for a new free surface elevation $S$ and we perform remeshing after each free-surface evolution computation, except for the Runge-Kutta time discretization option, where we perform more frequent mesh updates (i.e. remeshing of the 3D computational mesh). We added a description of the sequence of velocity computations, free–surface evolution and generating a new mesh / remeshing in the updated Section Model Chain including a flow chart illustrating this sequence.

We also updated the description of the meshing and remeshing procedure in general to clarify how a mesh update, i.e. remeshing, is performed, which has now been moved to Appendix A. The simulation framework offers two options for the generation of new meshes, i.e. mesh updates, both of them refer to remeshing. The standard approach (used for the glacier cases in the manuscript) is to use STL files for the generation of new meshes, i.e. mesh updates. In this case, the coordinates of the new surface elevation are used to set the vertical coordinate of the STL file gridpoints, which forms the surface of the desired 3D mesh. In a subsequent step, the interior volume is meshed using gmsh. Hence, this step is performed by remeshing and not just moving gridpoints vertically. In the second option, msh files are used to generate a new mesh, where the number of vertical layers has to be prescribed and this number of layers is then fixed. If a new mesh is generated, all gridpoints are moved vertically as a function of the overal shift in surface elevation. In this manner, mesh quality is ensured and no distortion of mesh cells is introduced. We have changed the formulation of this to clarify that we perform remeshing.

Regarding the comment on the need to perform mesh updates. When a new mesh is generated, this can be set by the user by defining the free surface evolution time step and should be chosen according to the problem at hand. Of course, a surface elevation change criterion can be introduced to define when a new mesh has to be generated, for example based on the maximum change in surface elevation. We mention for all the presented Tests within the respective Sections what interval is used for a mesh update and if the remeshing is performed using the STL or msh file option.

**Comment:** *page 6, line 21: I would suggest to use nodes instead of gridpoints.*

**Response:** Thanks for this comment, we now use nodes throughout.

**Comment:** *page 7, line 3: are the Stokes equations also solved in subdomain 2? Anyway, velocity computed on domain 2 should be~0 so that Eq. (3) should reduce to (4) even without the subdomain strategy?*

**Response:** Velocities are computed throughout the entire domain, i.e. subdomain 1 and 2, as, due to the shape of the cells, the margins of the domains do not have to be sufficiently smooth to represent high quality boundaries for the velocity computations. Hence, we perform velocity computations for the entire mesh. For regions where only the artificial ice layer is present (subdomain 2), velocities are indeed extremely small, however they are still not zero and show variations due to the variations in surface slope of the terrain topography, so we set them to zero.

**Comment:** *page 7, line 6: can you explain what you mean by a "velocity-dependent" buffer zone"?*

**Response:** For the free–surface computations, ice can only advance due to ice flow where the full form of Eq. 3 is solved (this is the case for subdomain 1). Hence, in order to allow the glacier to advance also into previously ice-free regions, subdomain 1 has to be chosen accordingly. This is done by enlarging subdomain 1 by a 'velocity-dependent' buffer zone. This bufferzone is defined by the maximum velocity, minimum grid size and the full time step of surface evolution. In this manner, glacier advance due to ice flow into previously ice-free terrain is facilitated. We include additional information: "Subdomain 1 is enlarged by a buffer zone based on the maximum potential displacement of the glacier front (defined as a function of max. velocity, min. mesh size), to facilitate glacier advance into ice-free areas." This is now to be found in the Section Numerical Implementation, where the ice velocity computation is described.

**Comment:** *page 7, lines 8-13: an example of where you explain how you solve the free-surface equation under the constraint $S > B$ but just giving the technical details of the modules used, not really the method. All these materials all along the manuscript should be grouped in part 2.*

**Response:** We restructured the manuscript and added information about the method used to solve the free–surface evolution including the inequality constraint in Sections Mathematical Formulation and Numerical Implementation, see details in 1. General Response.

**Comment:** *part 4.3: if you are using a vertically extruded mesh or a completely unstructured mesh should be mentioned here. Also, the last sentence is not very clear (what is the difference between creating a new mesh and remeshing?). A "msh" file is a bit technical. My understanding is that the STL surface has been modified, then the "msh file is necessarily modified and therefore a new mesh has to be created. So this should be done each time the variable S has been modified? Also, how the previous solution(velocity, but also S) is interpolated on the new mesh should be discussed.*

**Response:** Thanks for this comment, a detailed description of the mesh generation procedure is given in Appendix A, also describing under which settings vertically extruded or fully unstructured meshes are used. We now call this Section Preprocessing, mesh generation and remeshing and also describe the mesh update, which refers to the process of remeshing once a new surface elevation is available. Detailed information on when remeshing is performed is now given in the updated Section Model Chain. Regarding the interpolation, the x and y coordinates of the 3D and 2D mesh remain constant throughout the computations (the only exception is in case of adaptive mesh refinement). Hence, an actual solution of S and also the velocities exists on the gridpoints. However, FEniCS provides a method to evaluate a Function (if defined on a continuous function space) at any point within the computational domain. For this purpose, first the cell where this point is located in is identified and then a linear combination of basis functions is evaluated at the given point within the respective cell (information provided in The FEniCS Tutorial, Langtangen and Logg (2016)).

**Comment:** *page 9, line 6: It seems that $S = S_0$ is more an initial condition than a Dirichlet boundary condition? It is only set at $t = 0$ not for any t?*

**Response:** In this test, at the domain boundaries, the surface elevation is set to the elevation of the initial surface, which is $0$ m in this case. In this manner we set a Dirichlet condition on the boundaries of the domain for any $t$.

**Comment:** *page 9, around equation (5): I would suggest to avoid mixing of $_0$ and $_{init}$? $S_0$, $A_0$ and $V_0$ would be more consistent, meaning the value at $t = 0$. Also, in (5) $h$ should write $h_0$, but to avoid new variable to be introduced $h + z(t)$ could be replaced by $S(t) = S_0 + (u_z + a)t$?*

**Response:** Thanks for this comment, we replaced $A_{init}$ and $V_{init}$ by $A^0$ and $V^0$ to be more consistent in terms of parameter naming.

**Comment:** *page 11, lines 1-5: what is the boundary condition at the bedrock for the Stokes?*

**Response:** We mention on page 7, line 25 (location in initial manuscript version) that we perform simulations assuming no sliding at the glacier-bedrock interface. For further clarity, we also included this now in Sect. 5.2: " For the velocity computations, the normal velocity component is set to zero at the boundaries, so that the domain boundaries act as walls for ice flow and we assume no sliding at the glacier bed."

**Comment:** *page 11, line 11: give the SMB that you have used such that the test can be reproduced by other users. In general, the exhaustive setups of these tests that aim to serve here as benchmarks should be given.*

**Response:** Thanks for this comment, the full test setup including the mass balance rate function is provided in the assets of this manuscript (zenodo repository) as well as on the linked github repository (evolve_glacier/tests/Glacier). The provided test cases include the mass balance function and the required initial meshes, so that the tests shown here can be reproduced. We also added a hint to this in Sect. 5.2.

**Comment:** *page 12, line 6: 5.2.1 -> 5.2?*

**Response:** Section 5.2 refers to the real-world glacier tests and this is divided into: Sect. 5.2.1 test with zero mass balance and

Sect. 5.2.2 test with elevation dependent mass balance rate, whereas Sect. 5.3 employs the same glacier geometry but random input fields for mass balance rate and velocities.

**Comment:** *page 12, line 9: specify that Eq. (B2) is given in Appendix 2*

**Response:** We now refer to Eq. 9 in the main text, as the equations are all listed in the main text now and not in the Appendix anymore.

**Comment:** *page 12? line 15: give the value for the computational time step. It seems from this sentence that the Stokes and the free-surface solver are not solved at each time step. This should be explained before in the manuscript. Regarding the advancing and retreat cases, I would suggest to discuss the two situations already in the introduction. At the end, you are presenting both situations which have their own difficulties in terms of numerics (constraint $S > B$ when retreating, front oscillation when advancing).*

**Response:** We perform computations of velocities and free–surface evolution iteratively. The period of time, where the free–surface evolves is referred to as surface evolution time step. For example in the case of the Crank-Nicholson time discretization this means that, for a surface evolution time step set to one year, a velocity field is computed and the free–surface evolves for one year using this velocity field. After the period of one year, a new mesh is generated using the resulting surface elevation, a new velocity field is computed and the free–surface evolution can start again. The time step used to compute the free–surface elevation itself is referred to computational time step, which in the case of the Crank-Nicholson scheme is derived using the CFL condition and a Courant number of 0.1 (see Sect. 3). This is different for the implicit Euler or Runge-Kutta scheme. We added a detailed description of the surface evolution time step and the computational time step and the sequence of velocity computations and free–surface evolution in the updated Section Model Chain. Following the comment, we now use the term time step instead of computational time step, as this has been defined properly in Section Model Chain.

We now present both situations, advance and development of oscillations, retreat where the constraint is affected, now in the Introduction by adjusting the text to read: "For evaluating our scheme, we propose a new set of free–surface evolution benchmarks that will be useful tests for other existing or future implementations. These benchmarks thoroughly test the implementation of the inequality constraint, by introducing negative mass balance conditions that strongly affect the constrained solver. In the case of steep advancing fronts that represent strong gradients in surface elevation, finite element methods are prone to develop spurious oscillations in the vicinity of these fronts (Bochev et al., 2004). Regarding this issue, we propose an idealized hill test and present a review of the following stabilization schemes: ...."

**Comment:** *page 12, line 22: how much the initial volume of the pyramid is a function of the mesh resolution? There is a dot over 6 in 0.376?*

**Response:** We included the relative errors in initial volume due to the coarser mesh resolution in Section 6.1.1. The initial pyramid volume is $V_0 = 0.37\dot{6} \text{ m}^3$, where the dot indicates that 6 is a periodic number.

**Comment:** *Figure 5: relative difference instead of absolute would be more pertinent as nevertheless the choice of these values has no real meaning?*

**Response:** We chose the absolute numbers, because it demonstrates the convergence of the absolute error with decreasing mesh size.

**Comment:** *Figure 6 (and others): I would suggest to avoid the grey background and also the frame around panels. I would also suggest to use letter (a, b, c...) to label the different panels of a given figure. "N = 1000 resolution" looks strange (for the resolution N=1000).*

**Response:** Thanks for this suggestion, we changed it to N=1000 and adjusted the figure caption to clearly refer to the respective panels. However we choose to keep the grey background for clarity.

**Comment:** *Figure 8: ice thickness would be more informative, or at least the initial S should be drawn on these two panels giving S?*

**Response:** We want to show the temporal evolution of the simulations, this is why we chose to show the differences to the initial surface elevation: $S^0 - S(t)$, which corresponds to a change in ice thickness as the bed elevation is constant. The initial surface elevation is given in the lower panels in brown.

**Comment:** *page 18, line 3: and at other places. Why computational time step as it has a physical meaning. So time step alone would be better.*

**Response:** We now use time step for the time step used to perform the free–surface evolution computation throughout the paper, following the added information on time stepping and coupling in Section Model Chain.

**Comment:** *page 19, line 28: again, are the velocity and free surface solved using different timesteps?*

**Response:** This depends on the choice of time discretization, we added information to clearly describe this in the updated Section Model Chain.

**Comment:** *page 21, line 16: are display?*

**Response:** Thanks, corrected to: "However, the results show spurious..."

**Comment:** *page 21, line 18: region (configuration(ii) (see Fig. 13 and Fig. 14), results -> region(configuration (ii), see Fig. 13 and Fig. 14), results*

**Response:** Done.

**Comment:** *part 7.2 misses an analysis if the oscillations are created from the free surface equation and/or the Stokes velocity solution in line of what has been shown in John et al.(2018a)? With a perfect velocity field, would one get surface*

[Figure]

**Figure 1.** Left panel: wrap of the 2D concentration field at the initial time step $t = 0$. In the right panel, the concentration field after $50$ s of transport is shown.

*oscillations? With a perfect free surface solver, would one get oscillations?*

**Response:** We tested a simple configuration where a step in a e.g. concentration of $C = 200$ is transported by a prescribed, constant translational velocity field of $3\,\mathrm{ms}^{-1}$ along the x-axis and a constant source term of $-1\,\mathrm{s}^{-1}$ is introduced. This step in concentration is represented by a constant input of concentration at the left boundary. For this simple test, a Crank-Nicholson time discretization and SUPG stabilisation is employed. An inequality constraint is introduced, to enforce no-negativity of concentration ($C \geq 0$) . Translated to our glaciological application, this configuration could represent an advancing front with constant prescribed velocity (also in the vertical) and a negative mass balance rate that affects the bed constraint. This simple test configuration is shown in Fig. 1, with the start configuration in the left panel and the concentration field after $50$ s of transport. Also when a constant velocity field is prescribed, spurious oscillations develop for sufficiently steep shock fronts, here the step in concentration. However, this simple test also shows that capabilities of the SNES solver for solving the constrained problem as the concentration does not fall below $0$ despite the negative source term. As solving a pure advection problem is inherently difficult to finite element methods (e. Bochev et al., 2004) and following the presented findings, we suggest that even with a perfect velocity field one has to expect the development of oscillations given a sufficiently steep front, or sharp layer. If the front is less steep, as for example the pyramid in the paper, no spurious oscillations develop for the presented setup. To our knowledge existing advection problem numerics for finite element methods all face problems for transportation of very steep fronts. Hence, if a theoretical free–surface solver would produce oscillations due to the approximated velocity field remains to

be tested.

**Comment:** *Eq. (A1): define $S_0$, $S_{mid}$ ($S_{mid}$ defined after B1 should be defined here)*

**Response:** Done, we defined everything now in Section Mathematical Formulation.

**Comment:** *Eq. (A3): what is h here (I guess not the same as in (5))? $h_K$ was used somewhere in the text*

**Response:** In Eq. A3, $u_h$ is the horizontal component of the 3D velocity field, as defined in Sect. 2. We state on page 26, line 4: "If not stated differently, all utilized parameters refer to the same quantities as in the manuscript."

**Comment:** *make title A1, A2, etc... similar*

**Response:** We restructured the paper.

**Comment:** *page 28, line 5: t, t+1 should be defined before*

**Response:** Done.

**3   Response to Anonymous Referee #2**

We thank Anonymous Referee #2 for his valuable comments on our manuscript. We addressed and replied to the general comments of Referee #2 in the General Response (Sect. 1) to both referees in the beginning of this document. This response addresses a discussion of the variational inequality, and how it is solved numerically and details on the methods used. Furthermore, it includes a response to the comment on refocusing the paper, it being quite descriptive at times and on listing all the relevant equations including the weak forms earlier in the paper.

In addition to the General Response, in the following Section we list the remaining specific comments of Referee #2, that have not already been addressed in the General Response (Sect. 1), as individual comments in order to reply point-by-point on how these specific points have been tackled in the updated manuscript.

**3.1   Specific Comments**

**Comment:** *As far as I could see, most of the test presented related to how accurately the (SUPG stabilized) mass-transport equation is solved. The test are useful and doing these and similar test is an essential part of the model-development phase. I did not see that the thickness constraint is mass conserving, and the discussion on page 23 suggested that it is in fact not. However, almost all of the tests and the associated discussions revolved around the stabilisation of the surface elevation equation and this mass-conserving aspect was not really addressed or analysed. I in fact doubt that the thickness constraint can be locally mass conserving for any finite time step. I never saw the details of the method, but if this is solved using PETSc as a constraint minimisation problem, then I suspect the corresponding Lagrange multipliers can be thought of as fictitious mass*

*sources.*

**Response:** The pyramid test case (Test A in Sect. 5.1.1) is specifically designed to test for mass conservation with a thickness constraint as we have presented an exact solution for this problem in the manuscript. However all the other tests are hard to interpret for mass conservation as no readily available exact solution exists. It is very important to note that constrained mass conservation problems (like glacier surface evolution in our case) are not optimization problems. We talk about non-linear complementarity problem (NCP) and variational inequality (VI) formulations in our context, nevertheless they do not come from a symmetric Jacobian or Hessian (e.g. Bueler, 2016). We have now explained the solution process in PETSc in sufficient detail in the manuscript (eqs. 14-16 in the manuscript) and cite more detailed descriptions (Benson and Munson, 2006). The reduced space method we use through PETSc does not utilize Lagrange multipliers, hence the concept of Lagrange multipliers as "fictitious mass sources" does not apply in our case. Nevertheless an even more comprehensive study of mass conservation in thickness constrained surface flows is subject to future studies and should be carried out as soon as resources allow.

**Comment:** *Equations A2-A4 are referred to as being on variational form, but I do not see a variational form there. Also, should $\tau$ not be inside the integral as it is element dependent and therefore spatially variable? Most of the discussion in the appendixes is presumably 'common knowledge'. I would list these equations as a part of the whole system, but is there any reason to have an appendices on Crank-Nicholson, Runge-Kutta and Backward Euler in a professional journal?*

**Response:** Thanks for this comment, we now included all the relevant equations in the Section Mathematical Formulation and there is no Appendix anymore. We also moved $\tau$ to be inside of the integral as it is mesh size dependent, thanks for this hint.

**References**

Balay, S., Gropp, W. D., McInnes, L. C., and Smith, B. F.: Efficient Management of Parallelism in Object Oriented Numerical Software Libraries, in: Modern Software Tools in Scientific Computing, edited by Arge, E., Bruaset, A. M., and Langtangen, H. P., pp. 163–202, Birkhäuser Press, 1997.

5   Balay, S., Abhyankar, S., Adams, M. F., Brown, J., Brune, P., Buschelman, K., Dalcin, L., Dener, A., Eijkhout, V., Gropp, W. D., Kaushik, D., Knepley, M. G., May, D. A., McInnes, L. C., Mills, R. T., Munson, T., Rupp, K., Sanan, P., Smith, B. F., Zampini, S., Zhang, H., and Zhang, H.: PETSc Web page, http://www.mcs.anl.gov/petsc, 2018a.

Balay, S., Abhyankar, S., Adams, M. F., Brown, J., Brune, P., Buschelman, K., Dalcin, L., Dener, A., Eijkhout, V., Gropp, W. D., Kaushik, D., Knepley, M. G., May, D. A., McInnes, L. C., Mills, R. T., Munson, T., Rupp, K., Sanan, P., Smith, B. F., Zampini, S., Zhang, H., and

10   Zhang, H.: PETSc Users Manual, Tech. Rep. ANL-95/11 - Revision 3.9, Argonne National Laboratory, http://www.mcs.anl.gov/petsc, 2018b.

Benson, S. J. and Munson, T. S.: Flexible complementarity solvers for large-scale applications, Optim. Method. Softw., 21, 155–168, 2006.

Bochev, P. B., Gunzburger, M. D., and Shadid, J. N.: Stability of the SUPG finite element method for transient advection–diffusion problems, Comput. Method. Appl. M., 193, 2301–2323, https://doi.org/10.1016/j.cma.2004.01.026, 2004.

15   Bueler, E.: Computing glacier geometry in nonlinear complementarity problem form, in: Conference: "14th Copper Mountain Conference on Iterative Methods", 2016.

Langtangen, H. P. and Logg, A.: Solving PDEs in Python: The FEniCS Tutorial I, vol. 1, Springer International Publishing, 2016.

---

## Author Response (AR2)

Anna Wirbel[1] and Alexander Helmut Jarosch[2]

[1]Department of Atmospheric and Cryospheric Sciences, University of Innsbruck, Innsbruck, Austria
[2]ThetaFrame Solutions, Hörfarterstrasse 14, Kufstein, Austria.

**Correspondence:** Anna Wirbel (Anna.Wirbel@uibk.ac.at)

**1  General Response**

We thank both referees for reviewing the revised manuscript and their helpful and constructive comments. This document contains a point by point reply to the referees' comments and a marked-up version of the revised manuscript where changes are highlighted. Additionally, we changed the order of Figs. 12, 13 and 14 so that configuration (i) initially ice-covered hill appears first.

**2  Response to Referee #2**

We thank Referee #2 for the interesting comments and further ideas as well as helpful text edits.

**Comment:** *The authors have responded to my full satisfaction to all my earlier comments. I still don't understand how they solve the inequality problem, but form eq 10 is seems they might be using what I know as the 'active method'. There is now discussion however on how the iteration over the active set is conduced. Anyhow, I think since the paper now has another title this is not a bit issue anymore. And there are many different ways of solving such problems.*

**Response:** We are delighted to read that the reviewer approves our response. Regarding the inequality problem, yes we do use an "active set method", as stated in the manuscript in Eq. (13) and we feel that our description of the solving routine is adequate. More details can be found in the referenced Benson and Munson (2006) publication.

**Comment:** *The paper fits nicely to the scope of the journal and it's great to be able to read and learn about various tactical modelling details that one would typically not get from papers on other journals.*

**Response:** We appreciate this comment of the reviewer as it reflects our perception of the presented material.

**Comment:** $nxn$ -> $n \times n$ *on page 4.*

**Response:** We changed this and now use $n \times n$.

**Comment:** *'harder test' on page 12, maybe replace with 'more stringent' test*

5  **Response:** We changed this to 'more stringent' test.

**Comment:** *Just a comment: With respect to applying elevation-dependent mass balance (page 12 and there about). I've always found that it helps if this can be done implicitly. That is the da/dz term is taken over to the left-hand side and one solves for velocities and thickness at the same time. (I understand that in a full Stokes situation this is going to be more difficult than*

10  *in the situations that I usually deal with which as the SSA equations.)*

**Response:** Thanks for this comment, we agree that this would be very interesting to include in future work.

**Comment:** *I'm not sure this is really needed at this stage. But because you have so many tests to describe it might have been a good idea to list them in a able, or give them some more descriptive names.*

15  **Response:** Thanks for this comment, we adjusted the names of the tests to be more descriptive: The new names are (with old names in brackets): Test Pyramid Translation (Test A), Test Swirling Flow (Test B), Test Zero Mass Balance (Test 1), Test Elevation-Dependent Mass Balance (Test 2), Test Random Input Fields (Random glacier test).

**Comment:** *Line 15 page 19. Rewrite first sentence. Maybe: Occasionally, for very steep advancing glacier fronts, unphys-*

20  *ical oscillations developed.*

**Response:** We changes this sentence to: "However, in performing tests for a diverse set of mass balance conditions, occasionally for very steep advancing glacier fronts, unphysical spurious oscillations developed, ..."

**Comment:** *Page 19: Are you sure the 'wave' is a kinematic wave? It's almost impossible for kinematic waves to develop*

25  *on glaciers unless mean slopes very steep. But maybe these are finite-amplitude effects. See for example, Vieli, G. J. M. C. L., and Gudmundsson, G. H. (2010). A numerical study of glacier advance over deforming till. Cryosphere, 4(3), 359–372. https://doi.org/10.5194/tc-4-359-2010.*

**Response:** Thanks for this comment, we agree that the term 'kinematic wave' is misleading in this case, as the wave is just initiated by the mass overburden, so we removed the term 'kinematic'.

**Comment:** *I was just wondering if your SUPG stability parameter $tau = h/2u$ should possible have been selected as $1/tau = 1/taut + 1/taus$; where $taut = dt/2$ and $taus = h/2v$. This gives the limit $tau- > 0$ with $dt- > 0$ as is should be. However, I suspect that this might not actually help and that you would only get rid of the oscillations using some sort of forward/positive-backward/negative diffusion method.*

35  **Response:** Thank you for this comment. Your limit analysis is correct and we could re-write the SUPG stability term as suggested. We chose the one presented as it is often found in literature. Either way we agree that only a more complex diffusion method would get rid of the oscillations, which is planned for future work on this subject.

**3   Response to Anonymous Referee #1**

5   We thank Anonymous Referee #1 for the valuable comments on the revised manuscript and the edits regarding the notation used for the equations.

**3.1   General Comments**

**Comment:** *First, I think the paper could be improved by adding a systematic comparison (quality of the solution, cpu) of the 3 presented stabilisation methods for all examples. It is mentioned that all are done with the Crank–Nicholson method (page*
10   *11 line 1), but it seems it is not the case (discussion in 6.2.2).*
**Response:** Thanks for this comment. It is correct that the Crank-Nicholson time discretization is used for all simulation framework tests summarized in Sect. 5. A systematic comparison of the different stabilisation methods presented (all spatial stabilisation methods and time discretization schemes) is provided in Sect. 7.. This comparison is performed for the 'glacierized hill' test, where due to the development of extremely strong gradients in surface elevation, the occurrence of spurious
15   oscillations is most likely. So for this stringent test setup, we tested all the presented stabilisation methods and compared them in Figs. 11, 12, 13 and 14. In the tests presented in Sect. 5, we use the 'default' setup, Crank-Nicholson and SUPG method and present the results. In Test 2 (now: Test Elevation-Dependent Mass Balance) the 'default' setting does not guarantee stable results over the entire simulation period of $100$ years. However, if after $80$ years, the Runge-Kutta time discretization is used, stability is guaranteed. With this example, we want to point out the importance of testing and choosing the best working scheme
20   for the problem at hand. We hope to facilitate this choice by providing the option to chose from a set of spatial stabilisation methods and time discretizations and by illustrating and discussing the results of the tests presented in Sect. 7 and by listing the particular strengths and weaknesses of the different stabilisation methods, we intend to provide some guidance on how to chose a suitable stabilisation method.

25   **Comment:** *Second, the oscillation issue is not really shown in the main text (only in Fig 11 and then 12 and 13). Would it be possible to show on the other examples this issue with dedicated plots? Also, clearer conclusions should be drawn on which method(s) should be preferred?*
**Response:** In the main text, the simulation results are shown for different points in time of the simulation. Throughout the entire simulation period, in the case of the benchmark tests, no significant spurious oscillations develop. In the case of the glacier
30   simulations, by choosing appropriate stabilisation methods (as described in the main text), we could inhibit the development of significant spurious oscillations. In the case of Test Elevation-Dependent Mass Balance (former Test 2), this required switching to a Runge-Kutta time discretization for example. As demonstrated for the three different glacier tests, different problems can

be addressed by the application of different stabilisation methods. In a situation where gradients in the surface elevation remain smooth, a Crank-Nicholson time stepping might be fully sufficient, whereas in a fast-flowing, advancing glacier scenario, additional stabilisation will be required. This is what we demonstrate with the different glacier scenarios and the extended discussion in Sect. 8. Regarding a clearer conclusion, we state in Sect. 8: "This example illustrates that appropriate methods have to be chosen according to the problem at hand, as e.g. for complex situations like steep advancing fronts, a tighter coupling between velocity computations and free–surface evolution is favourable". It is our conclusion that that a preferable method can not be stated in general and that, as we write, the most suitable method needs to be determined by the "user" for a specific task at hand, balancing numerical accuracy and computational cost.

**3.2   Specific Comments**

**Comment:**   *Eq. (2): again I don't see why this equation is needed for what follow. At least, it should be given in term of stress to avoid an incomplete definition of the effective viscosity*

**Response:** We agree that it is not strictly needed here and removed Eq. 2.

**Comment:**   *page 4, line 11: Eq. (4) instead of (3)?*

**Response:** Thanks, we changed this to refer to Eq. 3.

**Comment:**   *In Eq. (5): put v always before (or after) the initial terms*

**Response:** Done.

**Comment:**   *In Eq. (8): not sure ":=" is needed ("=" is enough). Define what is F in (8). Is it a scalar or a vector?*

**Response:** We agree that '=' is sufficient. We have now simplified our notation and introduce two scalar functions, one for the NCP and one for the VI as these functions are not necessarily the same. Both are scalar however. In Equation (7) no function is required to state the system of equations and for simplicity we use matrix notation now.

**Comment:**   *page 4, line 19: $n \times n$ not $(nxn)$*

**Response:** Done.

**Comment:**   *In Eq. (9): is operator F the same as in (8) (so why in bold?)?*

**Response:** Not necessarily. We have clearly defined the used functions now.

**Comment:**   *In Eq. (11): "with" should not be in maths*

**Response:** Done.

**Comment:** *In Eq. (12): there is a missing integral sign?*

**Response:** Correct, we introduced the integral.

**Comment:** *After Eq. (13) should be mentioned how $F_{stab}$ is added to the functional?*

**Response:** We now write: "These are implemented by using the respective formulation of the stabilisation term $F_{stab}$ , which forms an additive term to the base functional. "

**Comment:** *page 5, line 15: in terms "of" stabilisation*

**Response:** Done.

**Comment:** *In Eq. (16) the notation should be given*

**Response:** Eq. (16) has been removed as it does not add information to the casual reader. If more in–depth information is required the reader is referred to Benson and Munson (2006).

**Comment:** *page 8, line 27: $S^1 = S^0 + \Delta t$ is not homogeneous. There is a velocity missing in front of $\Delta t$ to have a displacement*

**Response:** Thanks for this comment, we adjusted this to refer to the correct equation (Eq. 4).

**Comment:** *7.1.1: there is no 7.1.2*

**Response:** Thanks we changed this, now this became Sect. 7.2.

**References**

[revised manuscript text omitted]